# AlgoFormer: An Efficient Transformer Framework with Algorithmic Structures

**Yihang Gao**[1]* **Chuanyang Zheng**[2]* **Enze Xie**[3] **Han Shi**[3] **Tianyang Hu**[1] **Yu Li**[2]
**Michael Ng**[4] **Zhenguo Li**[3] **Zhaoqiang Liu**[5]†

{gaoyh,t.hu}@nus.edu.sg    {cyzheng21,liyu}@cse.cuhk.edu.hk    {xieenze}@connenct.hku.hk
{shi.ha,li.zhenguo}@huawei.com    michael-ng@hkbu.edu.hk    zqliu12@gmail.com

[1]National University of Singapore  [2]Chinese University of Hong Kong  [3]Huawei Noah's Ark Lab  [4]Hong Kong Baptist University [5]University of Electronic Science and Technology of China

**Reviewed on OpenReview:** `https://openreview.net/forum?id=oYP2Pd5aQt`

## Abstract

Besides natural language processing, transformers exhibit extraordinary performance in solving broader applications, including scientific computing and computer vision. Previous works try to explain this from the expressive power and capability perspectives that standard transformers are capable of performing some algorithms. To empower transformers with algorithmic capabilities and motivated by the recently proposed looped transformer (Yang et al., 2024; Giannou et al., 2023), we design a novel transformer framework, dubbed Algorithm Transformer (abbreviated as AlgoFormer). We provide an insight that efficient transformer architectures can be designed by leveraging prior knowledge of tasks and the underlying structure of potential algorithms. Compared with the standard transformer and vanilla looped transformer, the proposed AlgoFormer can perform efficiently in algorithm representation in some specific tasks. In particular, inspired by the structure of human-designed learning algorithms, our transformer framework consists of a pre-transformer that is responsible for task preprocessing, a looped transformer for iterative optimization algorithms, and a post-transformer for producing the desired results after post-processing. We provide theoretical evidence of the expressive power of the AlgoFormer in solving some challenging problems, mirroring human-designed algorithms. Furthermore, some theoretical and empirical results are presented to show that the designed transformer has the potential to perform algorithm representation and learning. Experimental results demonstrate the empirical superiority of the proposed transformer in that it outperforms the standard transformer and vanilla looped transformer in some specific tasks. An extensive experiment on real language tasks (e.g., neural machine translation of German and English, and text classification) further validates the expressiveness and effectiveness of AlgoFormer.

## 1 Introduction

The emergence of the transformer architecture (Vaswani et al., 2017) marks the onset of a new era in natural language processing. Transformer-based large language models (LLMs), such as BERT (Devlin et al., 2019) and GPT-3 (Brown et al., 2020), revolutionized impactful language-centric applications, including language translation (Vaswani et al., 2017; Raffel et al., 2020), text completion/generation (Radford et al., 2019; Brown et al., 2020), sentiment analysis (Devlin et al., 2019), and mathematical reasoning (Imani et al., 2023; Yu et al., 2024). Beyond the initial surge in LLMs, these transformer-based models have found extensive applications in diverse domains such as computer vision (Dosovitskiy et al., 2021), time series (Li et al., 2019),

---

*Equal contribution.
†Corresponding author.

bioinformatics (Zhang et al., 2023b), and addressing various physical problems (Cao, 2021). While many studies have concentrated on employing transformer-based models to tackle challenging real-world tasks, yielding superior performances compared to earlier models, the mathematical understanding of transformers remains incomplete.

Garg et al. (2022) empirically investigate the performance of transformers in in-context learning, where the input tokens are input-label pairs generated from classical machine learning models, e.g., (sparse) linear regression and decision tree. They find that transformers can perform comparably as standard human-designed machine learning algorithms. Some subsequent works try to explain the phenomenon. Akyürek et al. (2023) characterize decoder-based transformer as employing stochastic gradient descent for linear regression. Bai et al. (2023) demonstrate that transformers can address statistical learning problems and employ algorithm selection, such as ridge regression, Lasso, and classification problems. Zhang et al. (2023a) and Huang et al. (2023) simplify the transformer model with reduced active parameters, yet reveal that the simplified transformer retains sufficient expressiveness for in-context linear regression problems. In Ahn et al. (2023), transformers are extended to implement preconditioned gradient descent. The looped transformer is proposed in Giannou et al. (2023), and is shown to have the potential to perform basic operations (e.g., addition and multiplication), as well as implicitly learn iterative algorithms Yang et al. (2024). More related and interesting studies can be found in Huang et al. (2023); Von Oswald et al. (2023); Mahankali et al. (2024).

In this paper, inspired by the recently proposed looped transformer (Yang et al., 2024; Giannou et al., 2023), we propose a novel transformer framework, which we refer to as AlgoFormer, and strictly enforce it as an algorithm learner by the structure regularization on its architecture. The transformer framework consists of three modules (sub-transformers), i.e., the pre-, looped, and post-transformers, designed to perform distinct roles. The pre-transformer is responsible for preprocessing the input data, and formulating it into some mathematical problems. The looped transformer acts as an iterative algorithm in solving the hidden problems. Finally, the post-transformer handles suitable postprocessing to produce the desired results. In contrast to standard transformers, the AlgoFormer is more likely to implement algorithms, due to its algorithmic structures shown in Figure 1.

## 2  Motivation and Proposed Method

In this section, we mainly discuss the construction and intuition of the AlgoFormer. Its advantages over the standard transformer is then conveyed. Before going into details, we first elaborate the mathematical definition of transformer layers.

### 2.1  Preliminaries

A one-layer transformer is mathematically formulated as:

$$
\begin{aligned}
\text{Attn}\left(\boldsymbol{X}\right) &= \boldsymbol{X} + \sum_{i=1}^{h} \boldsymbol{W}_V^{(i)} \boldsymbol{X} \cdot \text{softmax}\left(\boldsymbol{X}^\top \boldsymbol{W}_K^{(i)\top} \boldsymbol{W}_Q^{(i)} \boldsymbol{X}\right), \\
\text{TF}\left(\boldsymbol{X}\right) &= \text{Attn}\left(\boldsymbol{X}\right) + \boldsymbol{W}_2 \text{ReLU}\left(\boldsymbol{W}_1 \text{Attn}\left(\boldsymbol{X}\right) + \boldsymbol{b}_1\right) + \boldsymbol{b}_2,
\end{aligned}
\tag{1}
$$

where $\boldsymbol{X} \in \mathbb{R}^{D \times N}$ is the input tokens; $h$ is the number of heads; $\{\boldsymbol{W}_V^{(i)}, \boldsymbol{W}_K^{(i)}, \boldsymbol{W}_Q^{(i)}\}$ denote value, key and query matrices at $i$-th head, respectively; $\{\boldsymbol{W}_2, \boldsymbol{W}_1, \boldsymbol{b}_2, \boldsymbol{b}_1\}$ are parameters of the shallow feed-forward ReLU neural network. The attention layer with softmax activation function mostly exchanges information between different tokens by the attention mechanism. Subsequently, the feed-forward ReLU neural network applies nonlinear transformations to each token vector and extracts more complicated and versatile representations.

### 2.2  Algorithmic Structures of Transformers

As discussed in the introduction, rather than simply interpreting it as an implicit function approximator, the transformer may in-context execute some implicit algorithms learned from training data. However, it is still unverified whether the standard multi-layer transformer is exactly performing algorithms.

**AlgoFormer**. As shown in the green part of Figure 1, vanilla looped transformers (Yang et al., 2024; Giannou et al., 2023) admit the same structure as iterative algorithms. However, real applications are usually

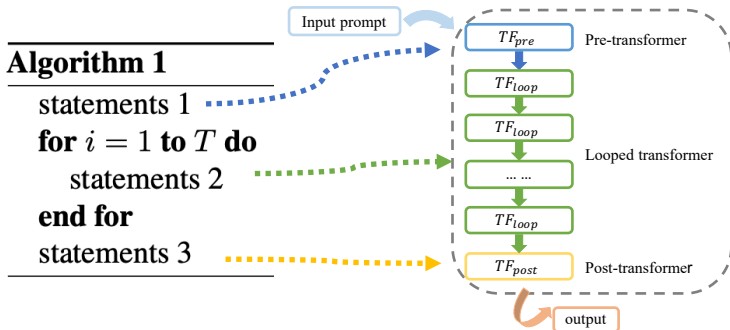

Figure 1: Algorithmic structure of the AlgoFormer. Here, $\text{TF}_{\text{pre}}$, $\text{TF}_{\text{loop}}$, and $\text{TF}_{\text{post}}$ are multi-layer transformers; "statements" represent some fundamental operations in classical algorithms.

much more complicated. For example, given a task with data pairs, a well-trained researcher may first pre-process the data under some prior knowledge, and then formulate a mathematical (optimization) problem. Following that, some designed solvers, usually iterative algorithms, are performed. Finally, the desired results are obtained after further post-processing. The designed AlgoFormer (Algorithm Transformer), visualized in Figure 1, enjoys the same structure as wide classes of algorithms. Specifically, we separate the transformer framework into three parts, i.e., pre-transformer $\text{TF}_{\text{pre}}$, looped transformer $\text{TF}_{\text{loop}}$, and post-transformer $\text{TF}_{\text{post}}$. Here, those three sub-transformers are standard multi-layer transformers in Equation (1). Given the input token vectors $\boldsymbol{X}$ and the number of iteration steps $T$, the output admits:

$$\text{TF}_{\text{post}} \underbrace{\left(\text{TF}_{\text{loop}} \left(\cdots \text{TF}_{\text{loop}}\right.\right.}_{T \text{ iterations}} \left(\text{TF}_{\text{pre}}(\boldsymbol{X})\right)\cdots). \tag{2}$$

The looped transformer layers ($\text{TF}_{\text{loop}}$) share the same set of weights, as they perform identical computations during each iteration. In contrast, the pre-transformer ($\text{TF}_{\text{pre}}$) and post-transformer ($\text{TF}_{\text{post}}$) utilize distinct weights and architectures to handle their specific computational roles. The hyperparameters for each transformer module (e.g., number of heads, layers, and hidden dimensions) are configured based on prior knowledge of the computational complexity of the target algorithms. Although we present AlgoFormer as in Equation (2), which consists of three modules, the key insight here is that the AlgoFormer can be designed flexibly based on the prior knowledge of the algorithm structure for the given task. Importantly, we do not restrict AlgoFormer to the specific form outlined in Equation (2). Compared with standard transformers, the AlgoFormer acts more as the algorithm learner, by strictly regularizing the loop structure. In contrast to Giannou et al. (2023) and Yang et al. (2024), which primarily focus on tasks solvable by iterative algorithms, our approach introduces additional transformer modules (e.g., pre- and post-transformers) for processing. These components are crucial for addressing the processing needs of real-world applications. This design makes the AlgoFormer capable of representing more complex algorithms and solving challenging tasks more efficiently. Additionally, one of the core insights of our work is that transformer architectures can be designed more efficiently, flexibly, and diversely by leveraging prior knowledge and the pre-defined structure of potential algorithms. This approach enables AlgoFormer to generalize across a broader range of applications while maintaining high efficiency and adaptability, such as representing algorithms involving nested loops, multiple loops, or multi-processing.

## 2.3 Training Strategy

Our training strategy builds upon the methodology introduced in Yang et al. (2024). Let $\boldsymbol{P}^i = [\boldsymbol{x}_1, f(\boldsymbol{x}_1), \cdots, \boldsymbol{x}_{i-1}, f(\boldsymbol{x}_{i-1}), \boldsymbol{x}_i]$ represents the input prompt for $1 \leq i \leq N$. We denote the AlgoFormer as $\text{TF}_{\text{Algo}}^t(\cdot; \boldsymbol{\Theta})$, where $f(\cdot)$ is a task-specific function that varies across different sequences and $t$ indicates the number of loops (iterations) in Equation 2, and $\boldsymbol{\Theta}$ represent the transformer parameters. Instead of evaluating the loss solely on $\text{TF}_{\text{Algo}}^T(\cdot; \boldsymbol{\Theta})$ with $T$ iterations, we minimize the expected loss over averaged iteration numbers:

$$\min_{\boldsymbol{\Theta}} \mathbb{E}_{\boldsymbol{P}} \left[ \frac{1}{T - T_0} \sum_{t=T_0}^{T} \frac{1}{N} \sum_{i=1}^{N} \left\| \text{TF}_{\text{Algo}}^t(\boldsymbol{P}^i; \boldsymbol{\Theta}) - f(\boldsymbol{x}_i) \right\|_2^2 \right], \tag{3}$$

where $T_0 = \max\{T - \Delta T, 0\}$ is the initial step for evaluating performance, $T$ is the maximal loop iterations during training, and $\Delta T$ is the number of loop iterations included in the loss. To stabilize the training, we adopt the moving average over loop iterations from $T_0$ to $T$ in the loss. Incorporating the loss from $T_0$ to $T$ ensures that the transformer module faithfully applies iterative algorithms and generalizes beyond the number of training loops. This approach helps enhance the transformer's capability for tasks requiring a varying number of iterations. Experimental results in Sections 5.2 and 5.3 further demonstrate that the transformer not only applies certain iterative algorithms but also generalizes well to longer loop iterations beyond those encountered during training.

## 3 Expressive Power

In this section, we theoretically show by construction that AlgoFormer is capable of solving some challenging tasks, akin to human-designed algorithms. The core idea is as follows. Initially, the pre-transformer undertakes the crucial task of preprocessing the input data, such as representation transformation. The looped transformer is responsible for iterative algorithms in optimization problems. Finally, it is ready to output the desired result by the post-transformer. Through the analysis of AlgoFormer's expressive power in addressing these tasks, we expect its potential to make contributions to the communities of scientific computing and machine learning. Throughout the section, we assume that the maximal number of data samples is $N$ and all sample observations (e.g., $\boldsymbol{x}_i$ and $\boldsymbol{y}_i$) are bounded.

### 3.1 Regression with Representation

We consider regression problems with representation, where the output behaves as a linear function of the input with a fixed representation function. Here, we adopt the $L$-layer MLPs with (leaky) ReLU activation function as the representation function $\Phi^*(\cdot)$. Specifically, we generate each in-context sample by first sampling the linear weight $\boldsymbol{A}$ from the prior $\mathcal{P}_A$, and then generating the input-label pair $\{(\boldsymbol{x}_i, \boldsymbol{y}_i)\}$ with $\boldsymbol{x}_i \in \mathbb{R}^d \sim \mathcal{P}_x$, $\boldsymbol{y}_i = \boldsymbol{A}\Phi^*(\boldsymbol{x}_i) + \boldsymbol{\epsilon}_i$ and $\boldsymbol{\epsilon}_i \sim \mathcal{N}(\boldsymbol{0}, \sigma^2 \boldsymbol{I})$. We aim to find the test label $\boldsymbol{y}_{\text{test}} := \boldsymbol{A}\Phi^*(\boldsymbol{x}_{\text{test}})$, given the in-context samples and test data $\{\boldsymbol{x}_1, \boldsymbol{y}_1, \cdots, \boldsymbol{x}_N, \boldsymbol{y}_N, \boldsymbol{x}_{\text{test}}\}$. Here, the weight matrix $A$ varies across different sequences of in-context samples but remains constant within a single sequence, and is learned in-context. The representation function $\Phi^*$, on the other hand, is fixed across all samples of sequences and is learned during training. A reliable solver is expected first to identify the representation function and transform the input data $\boldsymbol{x}$ to its representation $\Phi^*(\boldsymbol{x})$. Then it reduces to a regression problem, and some optimization algorithms are performed to find the weight matrix from in-context samples. Finally, it outputs the desired result $\boldsymbol{y}_{\text{test}}$ by applying transformations on the test data. We prove by construction that there exists an AlgoFormer that solves the task, akin to the human-designed reliable solver.

**Theorem 3.1.** *There exists a designed AlgoFormer with $TF_{pre}$ (an $(L+1)$-layer two-head transformer), $TF_{loop}$ (a one-layer two-head transformer), and $TF_{post}$ (a one-layer one-head transformer), that outputs $\boldsymbol{A}\Phi^*(\boldsymbol{x}_{test})$ from the input-label pairs $\{\boldsymbol{x}_1, \boldsymbol{y}_1, \cdots, \boldsymbol{x}_N, \boldsymbol{y}_N, \boldsymbol{x}_{test}\}$ by fitting the representation function and applying gradient descent for multi-variate regression. The emulation of each step is not exact, as there is some error introduced in each step. However, the error can be made arbitrarily close to zero by increasing the temperature of the softmax and adjusting another free parameter, neither of which affects the size of the network.*

**Remarks**. The detailed proof is available in Appendix A.1. Our construction of the transformer framework involves three distinct sub-transformers, each assigned specific responsibilities. The pre-transformer, characterized by identity attention, is dedicated to representation transformation through feed-forward neural networks. This stage reduces the task to a multivariate regression problem. Subsequently, the looped transformer operates in-context to determine the optimal weight, effectively acting as an iterative solver. Finally, the post-transformer is responsible for the post-processing and generate the desired result $\boldsymbol{A}\Phi^*(\boldsymbol{x}_{\text{test}})$. Here, the input prompt to the transformer is formulated as $\boldsymbol{P} = \begin{bmatrix} \boldsymbol{x}_1 & \boldsymbol{0} & \cdots & \boldsymbol{x}_N & \boldsymbol{0} & \boldsymbol{x}_{\text{test}} \\ \boldsymbol{0} & \boldsymbol{y}_1 & \cdots & \boldsymbol{0} & \boldsymbol{y}_N & \boldsymbol{0} \\ \boldsymbol{p}_1^x & \boldsymbol{p}_1^y & \cdots & \boldsymbol{p}_N^x & \boldsymbol{p}_N^y & \boldsymbol{p}_{N+1}^x \end{bmatrix}$, where $\boldsymbol{p}_i^x$, and $\boldsymbol{p}_i^y$ denote positional embeddings and will be specified in the proof. Due to the differing dimensions of the input $\boldsymbol{x}$ and its corresponding label $\boldsymbol{y}$, zero padding is incorporated to reshape them into vectors of the

same dimension. The structure of the prompt $\boldsymbol{P}$ aligns with similar formulations in previous works (Bai et al., 2023; Akyürek et al., 2023; Garg et al., 2022). For different input prompts $\boldsymbol{P}$, the hidden linear weights $\boldsymbol{A}$ are distinct but the representation function $\Phi^*(\cdot)$ is fixed. In comparison with the standard transformer adopted in Guo et al. (2024), which investigates similar tasks, the designed AlgoFormer has a significantly lower parameter size, making it closer to the envisioned human-designed algorithm. Notably, we construct the looped transformer to perform gradient descent for the multi-variate regression. However, the transformer exhibits remarkable versatility, as it has the capability to apply (ridge) regularized regression and more effective optimization algorithms beyond gradient descent. For more details, please refer to Section 4.

## 3.2 AR(q) with Representation

We consider the autoregressive model with representation. The dynamical (time series) system is generated by $\boldsymbol{x}_{t+1} = \boldsymbol{A}\Phi^*([\boldsymbol{x}_{t+1-q}, \cdots, \boldsymbol{x}_t]) + \boldsymbol{\epsilon}_t$, where $\Phi^*(\cdot)$ is a fixed representation function (e.g., we take the L-layer MLPs), and the weight $\boldsymbol{A}$ varies from different sequences but remains constant within a single sequence. Our goal is to learn $\Phi^*(\cdot)$ during training and to perform in-context learning of the weight matrix $\boldsymbol{A}$. In standard AR(q) (multivariate autoregressive) models, the representation function $\Phi^*(\cdot)$ is identity. Here, we investigate a more challenging situation in which the representation function is fixed but unknown. A well-behaved solver should first find the representation function and then translate it into a modified autoregressive model. With standard Gaussian priors on the white noise $\boldsymbol{\epsilon}_t$, the Bayesian estimator of the AR(q) model parameters admits $\arg\max_{\boldsymbol{A}} \prod_{t=1}^N f(\boldsymbol{x}_t|\boldsymbol{x}_{t-1}, \cdots, \boldsymbol{x}_{t-q}) = \arg\min_{\boldsymbol{A}} \sum_{t=1}^N \|\boldsymbol{x}_t - \boldsymbol{A}\Phi^*([\boldsymbol{x}_{t-q}, \cdots, \boldsymbol{x}_{t-1}])\|_2^2$, where $f(\boldsymbol{x}_t|\boldsymbol{x}_{t-1}, \cdots, \boldsymbol{x}_{t-q})$ is the conditional density function of $\boldsymbol{x}_t$, given previous $q$ observations. A practical solver initially identifies the representation function and transforms the input time series into its representation, denoted as $\Phi^*(\boldsymbol{x}_t)$. Then the problem is reduced to an autoregressive form. Similar to the previous subsection, we prove by construction that there exists a AlgoFormer, akin to human-designed algorithms, capable of effectively solving the given task.

**Theorem 3.2.** *There exists a designed AlgoFormer with $TF_{pre}$ (a one-layer q-head transformer with an $(L+1)$-layer one-head transformer), $TF_{loop}$ (a one-layer two-head transformer), and $TF_{post}$ (a one-layer one-head transformer), that predicts $\boldsymbol{x}_{N+1}$ from the data sequence $\{\boldsymbol{x}_1, \boldsymbol{x}_2, \cdots, \boldsymbol{x}_N\}$ by copying, transformation of the representation function and applying gradient descent for multi-variate regression. The emulation of each step is not exact, as there is some error introduced in each step. However, the error can be made arbitrarily close to zero by increasing the temperature of the softmax and adjusting another free parameter, neither of which affects the size of the network.*

**Remarks**. The detailed proof can be found in Appendix A.2. The technical details are similar to Theorem 3.1. The input prompt to the transformer is formulated as $\boldsymbol{P} = \begin{bmatrix} \boldsymbol{x}_1 & \boldsymbol{x}_2 & \cdots & \boldsymbol{x}_N \\ \boldsymbol{p}_1^x & \boldsymbol{p}_2^x & \cdots & \boldsymbol{p}_N^x \end{bmatrix}$, where $\boldsymbol{p}_i^x$ denote positional embeddings and will be specified in the proof. Additionally, the pre-transformer copies the feature from the previous $q$ tokens, utilizing $q$ heads for parallel processing.

## 3.3 Chain-of-Thought with MLPs

Chain-of-Thought (CoT) demonstrates exceptional performances in mathematical reasoning and text generation (Wei et al., 2022). The success of CoT has been theoretically explored, shedding light on its effectiveness in toy cases (Li et al., 2023) and on its computational complexity (Feng et al., 2023). In this subsection, we revisit the intriguing toy examples of CoT generated by leaky ReLU MLPs, denoted as CoT with MLPs, as discussed in Li et al. (2023). We begin by constructing an L-layer MLP with leaky ReLU activation. For an initial data point $\boldsymbol{x} \sim \mathcal{P}_x$, the CoT point $\boldsymbol{s}^\ell$ represents the output of the $\ell$-th layer of the MLP. Consequently, the CoT sequence $\{\boldsymbol{x}, \boldsymbol{s}^1, \cdots, \boldsymbol{s}^L\}$ is exactly generated as the output of each (hidden) layer of the MLP. The implicit $L$-layer MLP remains the same within a single sequence but varies across different sequences, and it is learned in-context. The target of CoT with MLPs problem is to find the next state $\hat{\boldsymbol{s}}^{\ell+1}$ based on the CoT samples $\{\boldsymbol{x}_1, \boldsymbol{s}_1^1, \cdots, \boldsymbol{s}_1^L, \boldsymbol{x}_2, \cdots, \boldsymbol{x}_N, \boldsymbol{s}_N^1, \cdots, \boldsymbol{s}_N^L, \boldsymbol{x}_{\text{test}}, \hat{\boldsymbol{s}}^1, \cdots, \hat{\boldsymbol{s}}^\ell\}$, where $\{\hat{\boldsymbol{s}}^1, \cdots, \hat{\boldsymbol{s}}^\ell\}$ denotes the CoT prompting of $\boldsymbol{x}_{\text{test}}$. We establish by construction in Theorem 3.3 that the AlgoFormer adeptly solves the CoT with MLPs problem, exhibiting a capability akin to human-designed algorithms.

**Theorem 3.3.** *There exists a designed AlgoFormer with $TF_{pre}$ (a seven-layer two-head transformer), $TF_{loop}$ (a one-layer two-head transformer), and $TF_{post}$ (a one-layer one-head transformer), that finds $\hat{\boldsymbol{s}}^{\ell+1}$ from samples $\{\boldsymbol{x}_1, \boldsymbol{s}_1^1, \cdots, \boldsymbol{s}_1^L, \boldsymbol{x}_2, \cdots, \boldsymbol{x}_N, \boldsymbol{s}_N^1, \cdots, \boldsymbol{s}_N^L, \boldsymbol{x}_{test}, \hat{\boldsymbol{s}}^1, \cdots, \hat{\boldsymbol{s}}^\ell\}$ by filtering and applying gradient descent for multi-variate regression. The emulation of each step is not exact, as there is some error introduced in each step. However, the error can be made arbitrarily close to zero by increasing the temperature of the softmax and adjusting another free parameter, neither of which affects the size of the network.*

**Remarks**. We put the proof in Appendix A.3. The pre-transformer first identifies the positional number $\ell$, and subsequently filters the input sequence into $\{\boldsymbol{s}_1^\ell, \boldsymbol{s}_1^{\ell+1}, \boldsymbol{s}_2^\ell, \boldsymbol{s}_2^{\ell+1}, \cdots, \boldsymbol{s}_N^\ell, \boldsymbol{s}_N^{\ell+1}, \hat{\boldsymbol{s}}^\ell\}$. This filtering transformation reduces the problem to a multi-variate regression problem. Compared with Li et al. (2023), where an assumption is made, we elaborate on the role of looped transformers in implementing gradient descent. While the CoT with MLPs may not be explicitly equivalent to CoT tasks in real applications, Theorem 3.3 somewhat implies the potential of the AlgoFormer in solving CoT-related problems.

## 4 Extensions and Further Analysis

In this section, we provide complementary insights to the results discussed in Section 3. Firstly, as discussed in the remark following Theorem 3.1, we construct the looped transformer that employs gradient descent to solve (regularized) multi-variate regression problems. However, in practical scenarios, the adoption of more efficient optimization algorithms is often preferred. Investigating the expressive power of transformers beyond gradient descent is both intriguing and appealing. As stated in Theorem 4.1, we demonstrate that the AlgoFormer can proficiently implement Newton's method for solving linear regression problems. Secondly, the definition in Equation 1 implies the encoder-based transformer. In practical applications, a decoder-based transformer with causal attention, as seen in models like GPT-2 (Radford et al., 2019), may also be favored. For completeness, it is also compelling to examine the behavior of decoder-based transformers in algorithmic learning. Our findings, presented in Theorem 4.2, reveal that the decoder-based AlgoFormer can also implement gradient descent in linear regression problems. The primary distinction lies in the fact that the decoder-based transformer utilizes previously observed data to evaluate the gradient, while the encoder-based transformer calculates the gradient based on the full data samples.

### 4.1 Beyond the Gradient Descent

Newton's (second-order) methods enjoy superlinear convergence under some mild conditions, outperforming gradient descent with linear convergence. This raises a natural question:

*Can the transformer implement algorithms beyond gradient descent, including higher-order optimization algorithms?*

In this section, we address this question by demonstrating that the designed AlgoFormer can also realize Newton's method in regression problems. Consider the linear regression problem given by:

$$\arg\min_{\boldsymbol{w}} \frac{1}{2N} \sum_{i=1}^{N} \left(\boldsymbol{w}^\top \boldsymbol{x}_i - y_i\right)^2. \tag{4}$$

Denote $\boldsymbol{X} = [\boldsymbol{x}_1, \boldsymbol{x}_2, \cdots, \boldsymbol{x}_N]^\top \in \mathbb{R}^{N \times d}$, $\boldsymbol{y} = [y_1, y_2, \cdots, y_N]^\top \in \mathbb{R}^{N \times 1}$ and $\boldsymbol{S} = \boldsymbol{X}^\top \boldsymbol{X}$. A typical Newton's method for linear regression problems follows the update scheme:

$$\boldsymbol{M}_0 = \alpha \boldsymbol{S}, \text{ where } \alpha \in \left(0, \frac{2}{\|\boldsymbol{S}\boldsymbol{S}^\top\|_2}\right], \quad \boldsymbol{M}_{k+1} = 2\boldsymbol{M}_k - \boldsymbol{M}_k \boldsymbol{S} \boldsymbol{M}_k, \quad \boldsymbol{w}_k^{\text{Newton}} = \boldsymbol{M}_k \boldsymbol{X}^\top \boldsymbol{y}. \tag{5}$$

As described in Söderström & Stewart (1974); Pan & Schreiber (1991), the above update scheme (Newton's method) enjoys superlinear convergence, in contrast to the linear convergence of gradient descent. The following theorem states that Newton's method in Equation 5 can be realized by the AlgoFormer.

**Theorem 4.1.** *There exists a designed AlgoFormer with $TF_{pre}$ (a one-layer two-head transformer), $TF_{loop}$ (a one-layer two-head transformer), and $TF_{post}$ (a two-layer two-head transformer), that implements Newton's method described by Equation 5 in solving regression problems. The emulation of each step is not exact,*

*as there is some error introduced in each step. However, the error can be made arbitrarily close to zero by increasing the temperature of the softmax and adjusting another free parameter, neither of which affects the size of the network.*

**Remarks**. The proof can be found in Appendix A.4. The pre-transformer performs preparative tasks, such as copying from neighboring tokens. The looped-transformer is responsible for updating and calculating $M_k x_i$ for each token $x_i$ at every step $k$. The post-transformer compute the final estimated weight $w_T^{\text{Newton}}$ and outputs the desired the results $w_T^{\text{Newton}\top} x_{\text{test}}$, where $T$ is the iteration number in Equation 2 and Equation 5. In a related study by Fu et al. (2023), similar topics are explored, indicating that transformers exactly perform higher-order optimization algorithms. However, our transformer architectures differ, and technical details are distinct.

### 4.2 Decoder-based Transformer

In the preceding analysis, the encoder-based AlgoFormer (with full attention) demonstrates its capability to solve problems by performing algorithms. Previous studies (Giannou et al., 2023; Bai et al., 2023; Zhang et al., 2023a; Huang et al., 2023; Ahn et al., 2023) also focus on the encoder-based models. We opted for an encoder-based transformer because full-batch data is available for estimating gradient and Hessian information. However, in practical applications, decoder-based models, like GPT-2, are sometimes more prevalent. In this subsection, we delve into the performance of the decoder-based model when executing iterative optimization algorithms, such as gradient descent, to solve regression problems.

We consider the linear regression problem in Equation 4. Due to the limitations of the decoder-based transformer, which can only access previous tokens, implementing iterative algorithms based on the entire batch data is not feasible. However, it is important to note that the current token in a decoder-based transformer can access data from all previous tokens. To predict the label $y_i$ based on the input prompt $P^i = [x_1, y_1, \cdots, x_i]$, the empirical loss for the linear weight at $x_i$ is given by

$$w^i \in \arg\min_{w} \mathcal{L}\left(w; P^i\right) := \frac{1}{2(i-1)} \sum_{j=1}^{i-1} \left(w^\top x_j - y_j\right)^2. \tag{6}$$

In essence, the linear weight is estimated using accessible data from the previous tokens, reflecting the restricted information available in the decoder-based transformer.

**Theorem 4.2.** *There exists a designed AlgoFormer with $TF_{pre}$ (a one-layer two-head transformer), $TF_{loop}$ (a one-layer two-head transformer), and $TF_{post}$ (a two-layer two-head transformer), that outputs $w_T^{i\top} x_i$ for each input data $x_i$, where $w_T^i$ comes from $\arg\min_{w} \mathcal{L}\left(w; P^i\right)$ after $T$ steps of gradient descent. The emulation of each step is not exact, as there is some error introduced in each step. However, the error can be made arbitrarily close to zero by increasing the temperature of the softmax and adjusting another free parameter, neither of which affects the size of the network.*

**Remarks**. The detailed proof is available in Appendix A.5. The technical details closely resemble those in Theorem 3.1, with the key distinction being that the decoder-based transformer can solely leverage data from previous tokens to determine the corresponding weight $w^i$. Our findings align with those in Guo et al. (2024), although our transformer architectures and attention mechanisms differ. In a related study by Akyürek et al. (2023), similar topics are explored, demonstrating that the decoder-based transformer performs single-sample stochastic gradient descent, while our results exhibit greater strength with $w^i \in \arg\min_{w} \mathcal{L}\left(w; P^i\right)$. The construction of a decoder-based transformer for representing Newton's method is more challenging. We left it as a potential topic for future investigation.

## 5 Experiments

In this section, we conduct a comprehensive empirical evaluation of the performance of AlgoFormer in tackling challenging tasks, specifically addressing regression with representation, AR(q) with representation, and CoT with MLPs, as outlined in Section 2. Additionally, we also implement AlgoFormer on the neural machine translation of German and English and AG News classification, demonstrating its expressiveness and effectiveness in real-world language tasks.

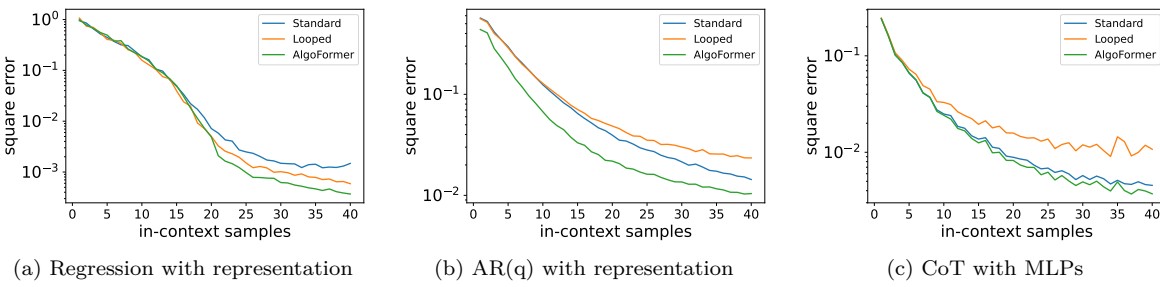

| (a) Regression with representation | (b) AR(q) with representation | (c) CoT with MLPs |

Figure 2: The validation error of trained models (the standard transformer, the vanilla looped transformer, and the AlgoFormer), assessed on regression with representation, AR(q) with representation, and CoT with MLPs tasks. By choosing suitable hyperparameters (i.e., we set $(T, \Delta T) = (20, 15)$), the AlgoFormer has significantly better performance than the standard transformer and the vanilla looped transformer on those tasks.

## 5.1 Experimental Settings and Hyperparameters

**Experimental settings**. In all experiments, we adopt the decoder-based AlgoFormer, standard transformer (GPT-2), and vanilla looped transformer Yang et al. (2024). For synthetic tasks, we utilize $N = 40$ in-context samples as input prompts and $d = 20$ dimensional vectors with $D = 256$ dimensional positional embeddings for all experiments. To ensure fairness in comparisons, all models are trained using the Adam optimizer, with a learning rate $\eta = 1e - 4$ and a total of 500K iterations to ensure convergence. The standard transformer is designed to have $L = 12$ layers while pre-, looped and post-transformers are all implemented in one-layer. The default setting for the AlgoFormer, as well as the vanilla looped transformer, involves setting $(T, \Delta T) = (20, 15)$. Here, the prompt formulation and the training loss in Equation (3) may slightly differ for different tasks. For example, in the AR(q) task, the input prompt is reformulated as $\boldsymbol{P}^i = [\boldsymbol{x}_1, \cdots, \boldsymbol{x}_{i-1}, \boldsymbol{x}_i]$ and $\boldsymbol{x}_{i+1} = f([\boldsymbol{x}_{i+1-q}, \cdots, \boldsymbol{x}_i])$ is the target for prediction. However, the training strategy can be easily transmitted to other tasks. Here, both the iteration numbers $T_0$ and $T$ are hyperparameters, which will be analyzed in the next subsection.

**Regression with representation**. In this task, we instantiate a 3-layer leaky ReLU MLPs, denoted as $\Phi^*(\cdot)$, which remains fixed across all tasks. The data generation process involves sampling a weight matrix $\boldsymbol{A} \in \mathbb{R}^{1 \times 20}$. Subsequently, input-label pairs $\{(\boldsymbol{x}_i, y_i)\}_{i=1}^{40}$ are generated, where $\boldsymbol{x}_i \sim \mathcal{N}(\boldsymbol{0}, \boldsymbol{I}_{20})$, $\epsilon_i \sim \mathcal{N}(0, \sigma^2)$ and $y_i = \boldsymbol{A}\Phi^*(\boldsymbol{x}_i) + \epsilon_i$. In Figure 2a, we specifically set $\sigma = 0$. Additionally, we explore the impact of different noise levels by considering $\sigma = 0.1$ and $\sigma = 1$. Here, our target is to learn the representation function $\Phi^*(\cdot)$ during training and to perform in-context learning of the weight matrix $\boldsymbol{A}$.

**AR(q) with representation**. For this task, we set $q = 3$ and employ a 3-layer leaky ReLU MLP denoted as $\Phi^*(\cdot)$, consistent across all instances. The representation function accepts a 60-dimensional vector as input and produces 20-dimensional feature vectors. The time series sequence $\{\boldsymbol{x}_t\}_{t=1}^N$ is generated by initially sampling $\boldsymbol{A} \sim \mathcal{N}(\boldsymbol{0}, \boldsymbol{I}_{20 \times 20})$. Then the sequence is auto-regressively determined, with $\boldsymbol{x}_{t+1} = \boldsymbol{A}\Phi^*([\boldsymbol{x}_{t+1-q}, \cdots, \boldsymbol{x}_t]) + \boldsymbol{\epsilon}_t$, where $\boldsymbol{\epsilon}_t \sim \mathcal{N}(\boldsymbol{0}, \boldsymbol{I}_{20})$. Here, our target is to learn the representation function $\Phi^*(\cdot)$ during training and to perform in-context learning of the weight matrix $\boldsymbol{A}$.

**CoT with MLPs**. In this example, we generate a 6-layer leaky ReLU MLP to serve as a CoT sequence generator, determining the length of CoT steps for each sample to be six. The CoT sequence, denoted as $\{\boldsymbol{x}, \boldsymbol{s}^1, \cdots, \boldsymbol{s}^L\}$ is generated by first sampling $\boldsymbol{x} \sim \mathcal{N}(\boldsymbol{0}, \boldsymbol{I}_{20})$, where $\boldsymbol{s}^\ell \in \mathbb{R}^{20}$ represents the intermediate state output from the $\ell$-th layer of the MLP. Here, our target is to perform in-context learning of the generator function (i.e., the 6-layer leaky ReLU MLP).

## 5.2 AlgoFormer Exhibits High Expressiveness

In this subsection, we conduct a comparative analysis of the AlgoFormer against the standard and vanilla looped transformers across challenging tasks, as outlined in Section 3. Figure 2 illustrates the validation error trends, showcasing a decrease with an increasing number of in-context samples, aligning with our intuition.

Crucially, the AlgoFormer consistently outperforms both the standard and the vanilla looped transformer across all tasks, highlighting its superior expressiveness in algorithm learning. Particularly in the CoT with MLPs task, both the AlgoFormer and the standard transformer significantly surpass the vanilla looped transformer. This further underscores the significance of preprocessing and postprocessing steps in handling complex real-world applications. The carefully designed algorithmic structure of the AlgoFormer emerges as an effective means of structural regularization, contributing to enhanced algorithm learning capabilities.

### 5.3 Impact of Hyperparameters

In this subsection, we conduct the empirical analysis of the impact of the hyperparameters on the AlgoFormer.

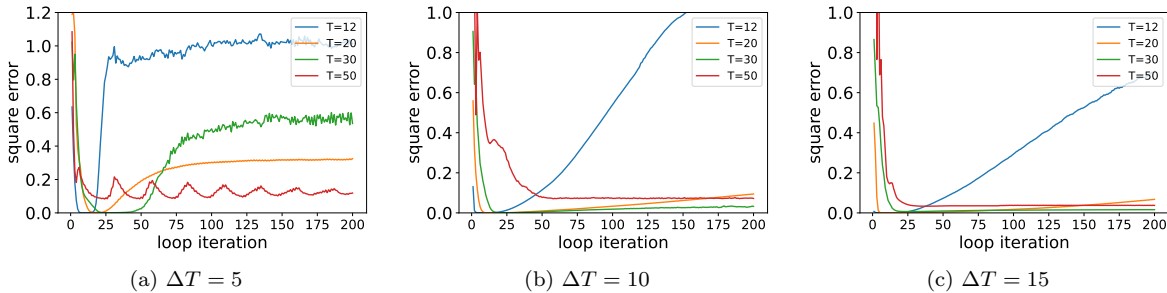

(a) $\Delta T = 5$        (b) $\Delta T = 10$        (c) $\Delta T = 15$

Figure 3: The validation error of trained models, evaluated on regression with representation task, with varying hyperparameters $T$ and $\Delta T$. The AlgoFormers are trained for $T$ loops, defined in Equation 3, and the evaluation focuses on square loss at longer iterations, where the number of loop iterations far exceeds $T$.

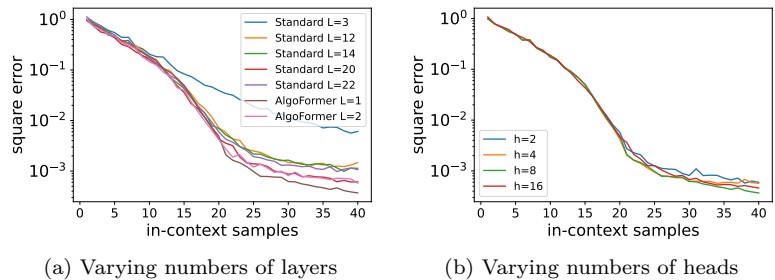

(a) Varying numbers of layers        (b) Varying numbers of heads

Figure 4: The validation error of trained models, evaluated on regression with representation task, with varying numbers of layers (denoted as $L$) and heads (denoted as $h$). In the context of AlgoFormer, the number of layers $L$ corresponds to the layers in the pre-, looped, and post-transformers, all of which are $L$-layer transformers. The AlgoFormers are trained with $(T, \Delta T) = (20, 15)$, defined in Equation 3.

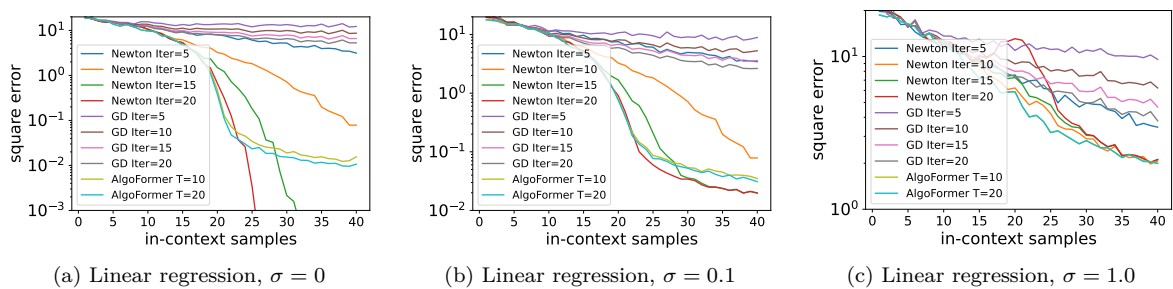

(a) Linear regression, $\sigma = 0$        (b) Linear regression, $\sigma = 0.1$        (c) Linear regression, $\sigma = 1.0$

Figure 5: The validation error of trained AlgoFormer models and the linear regression models optimized by gradient descent and Newton's method. The AlgoFormers are trained with $(T, \Delta T) = (20, 15)$ and $(T, \Delta T) = (10, 10)$, defined in Equation 3.

**Loop iterations**. We conduct comprehensive experiments on the AlgoFormer with varying loop numbers, on solving the regression with representation task. The results highlight the crucial role of both $T$ and $\Delta T$

in the performance of the AlgoFormer. It is observed that a larger $\Delta T$ contributes to the stable inference of transformers. Comparing Figure 3a with Figures 3b and 3c, it is evident that a larger $\Delta T$ enhances stable long-term inference. The number of loop iterations $T$ determines the model capacity in expressing algorithms. However, it is important to note that there exists a trade-off between the iteration numbers $(T, \Delta T)$ and computational costs. Larger $(T, \Delta T)$ certainly increases model capacity but also leads to higher computational costs and challenges in model training, as reflected in Figure 3.

**Number of heads and layers**. In our experiments on the AlgoFormer, we vary the numbers of heads and layers in $\text{TF}_{\text{loop}}$ while addressing the regression with representation task. The results reveal a consistent trend that an increase in both the number of heads and layers leads to lower errors. This aligns with our intuitive understanding, as transformers with greater numbers of heads and layers exhibit enhanced expressiveness. However, a noteworthy observation is that 4-layer and 16-head transformers may exhibit suboptimal performance, possibly due to increased optimization challenges during model training. This finding underscores the importance of carefully selecting the model size, as a larger model, while offering higher expressiveness, may present additional training difficulties. The visualized results are shown in Figure 4. Moreover, compared with the standard transformer, even with the same number of layers, the AlgoFormer exhibits better performance in those tasks, mainly due to the introduced algorithmic structure. This finding highlights the role of the structure regularization of the model. Therefore, we have reasons to believe that the good performance of the AlgoFormer not only comes from the higher expressiveness with deeper layers but also from the structure regularization of model architecture, which facilitates easier training and good generalization.

In Figure 3, we also provide insights into the behavior of AlgoFormer during inference. Although the AlgoFormer is trained with T=20 loop iterations, the inference is conducted with significantly longer loop iterations (e.g., T=200), As T increases beyond the training length, AlgoFormer demonstrates improved performance and eventually stabilizes (i.e., converges). This behavior suggests that the inner looped transformers are effectively performing iterative computations that converge to a stable solution.

### 5.4 AlgoFormer and Human-Designed Algorithms

In this subsection, we compare the AlgoFormer with Newton's method and gradient descent in solving linear regression problems. We adopt the same default hyperparameters, with their selection grounded in a comprehensive grid search.

As illustrated in Figure 5, we observe that in the noiseless case, the AlgoFormer outperforms both Newton's method and gradient descent in the beginning stages. However, Newton's method suddenly achieves nearly zero loss ( machine precision) later on, benefiting from its superlinear convergence. In contrast, our method maintains an error level around $1e - 3$. With increasing noise levels, both Newton's method and gradient descent converge slowly, while our method exhibits better performance.

Several aspects contribute to this phenomenon. Firstly, in the noiseless case, Newton's method can precisely recover the weights through the linear regression objective in Equation 6, capitalizing on its superlinear convergence. On the other hand, the AlgoFormer operates as a black-box, trained from finite data. While we demonstrate good model expressiveness, the final generalization error of the trained transformer results from the model's expressiveness, the finite number of training samples, and the optimization error. Despite exhibiting high expressiveness, the trained AlgoFormer cannot eliminate the last two errors entirely. This observation resonates with similar findings in solving partial differential equations (Raissi et al., 2019) and large linear systems (Gu & Ng, 2023) using deep learning models.

Secondly, with larger noise levels, Newton's method shows suboptimal results. This is partly due to the inclusion of noise, which slows down the convergence rate, and Newton's method experiences convergence challenges when moving away from the local solution. In terms of global convergence, the AlgoFormer demonstrates superior performance compared to Newton's method.

Human-designed algorithms, backed by problem priors and precise computation, achieve irreplaceable performance. It's important to note that deep learning models, including transformers, are specifically designed for solving black-box tasks where there is limited prior knowledge but sufficient observation samples. We

expect that transformers, with their substantial expressiveness, hold the potential to contribute to designing implicit algorithms in solving scientific computing tasks.

## 5.5 Applications to Language Tasks

In this subsection, we extend the evaluation of the proposed AlgoFormer to real-world language tasks, complementing its performance in real applications. Specifically, we focus on Neural Machine Translation using the IWSLT 2015 German-English dataset. The experimental setup includes a standard Transformer with 12 layers, 8 attention heads, a feature dimension of 256, and a learning rate of 5e-5. The pre-, looped, and post-transformers are all implemented as single layers, with $T$ set to 10 and $\Delta T$ to 10 (see the hyperparameters in Equation 3 for training). The input German text is treated as a prefix, and the output English text is generated autoregressively using decoder-based transformers for all three models. The translation performance is evaluated using cross-entropy loss, where lower values indicate better results. Additionally, BLEU (Bilingual Evaluation Understudy) is a widely used metric for measuring the quality of translation tasks, with higher scores indicating better performance. The results are presented in the table below:

| Model | Standard | Looped | AlgoFormer |
|---|---|---|---|
| **Cross entropy** | 4.99 | 4.73 | **4.61** |
| **BLUE** | 9.30 | 10.56 | **14.72** |

Table 1: Quality of Transformer models on machine translation.

We also implement the proposed AlgoFormer on the text classification task using various datasets (AG News, IMDB, DBPedia, Yelp Review, and Yahoo News) to evaluate its performance on a real-world language application. The experimental setup includes a standard Transformer with 12 layers, 1 attention head, a feature dimension of 32, and a learning rate of 1e-3. The pre-, looped, and post-transformers are all implemented as single layers with one attention head, with $(T, \Delta T) = (10, 10)$. The input news text data is processed using encoder-based transformers, and the output classification is generated. Classification accuracy, where higher accuracy indicates better performance, is used as the evaluation metric. The results are summarized in the table below:

| Model | Standard | Looped | AlgoFormer |
|---|---|---|---|
| **AG News** | 92.07 | 91.04 | **97.92** |
| **IMDB** | 75.00 | **77.50** | **77.50** |
| **DBPedia** | **99.21** | 99.11 | 98.21 |
| **Yelp Review** | **66.25** | 53.57 | 57.50 |
| **Yahoo News** | 73.96 | 69.79 | **81.25** |

Table 2: Accuracy (%) of Transformer models on text classification across different datasets.

As shown in the results, AlgoFormer outperforms and performs comparably with the standard Transformer and the vanilla looped Transformer in the Neural Machine Translation and Text Classification tasks, suggesting that conventional models may have inherent redundancies. This aligns with recent findings on the redundancy in large language models (Chen et al., 2024; Frantar & Alistarh, 2023; Xia et al., 2024). These preliminary results indicate the potential of AlgoFormer, which is designed as an algorithmic conductor, in real-world language tasks.

## 5.6 Computation Comparison

The additional computational costs and complexity introduced by AlgoFormer over the standard transformer and the vanilla looped transformer depend on the specific structure of the AlgoFormer framework. In our experimental settings, these costs are negligible. Compared to the standard transformer, the total computation overhead is primarily determined by the loop iterations $T$ and the number of loops $\Delta T$ included in the training loss. Since $T$ in our experiments is comparable to the number of layers in the standard transformer, the dominant additional computation comes from $\Delta T$. This overhead can be mitigated by using smaller batch sizes, ensuring that the proposed method does not introduce significant computational

overhead compared to the standard transformer. The main complexity lies in the training strategy outlined in Equation 3, which ensures stable training.

When compared to the vanilla looped transformer, the computational overhead comes from the pre- and post-transformer modules. However, if these modules are of similar size to the looped transformer, the additional computational cost is negligible, especially when the loop iteration $T$ is much greater than the number of layers in these modules. For instance, in our experiments, we set the pre-, looped-, and post-transformer layers to 1, with $T = 20$ during training. In this scenario, the additional computation introduced by the pre- and post-transformer modules constitutes approximately 1/10 of the total computation. During inference, where $T$ is typically much larger than during training (e.g., T=200 loop iterations in Figure 3), the additional computational overhead is further reduced to about 1/100.

## 6 Conclusion and Discussion

In this paper, we introduce a novel AlgoFormer, an algorithm learner designed from the looped transformer, distinguished by its algorithmic structures. We provide an insight that the efficient transformer framework can be designed by considering the prior knowledge of the task and the structure of potential algorithms. Comprising three sub-transformers, each playing a distinct role in algorithm learning, the AlgoFormer demonstrates expressiveness and efficiency while maintaining a low parameter size. Theoretical analysis establishes that the AlgoFormer can tackle challenging in-context learning tasks, mirroring human-designed algorithms. Our experiments further validate our claim, showing that the proposed transformer outperforms both the standard transformer and the vanilla looped transformer in specific algorithm learning and real-world language tasks.

While the proposed AlgoFormer framework offers flexibility and efficiency, several potential limitations and challenges warrant discussion.

First, the dependence on prior knowledge for designing concrete architecture. The design of the AlgoFormer framework relies heavily on prior knowledge of tasks and the structure of potential algorithms. In scientific computing tasks, where problems are often well-formulated and informed by domain knowledge, this reliance can be an advantage, enabling the design of task-specific, efficient architectures. However, for real-world language tasks, the abstract and unstructured nature of the problems often limits the availability of such prior knowledge. This can make it challenging to design an optimal AlgoFormer architecture for these tasks. While the insights of AlgoFormer can significantly benefit the scientific computing community, extending its design principles to language tasks may require more advanced techniques, such as automated architecture search or meta-learning, to determine suitable architectures without extensive prior knowledge.

Second, the training complexity of AlgoFormer with more complicated algorithmic structures. Although incorporating more complex algorithmic structures into AlgoFormer, such as nested looped transformers, enhances its expressiveness, training AlgoFormer can become more challenging. It inevitably introduces additional computational costs and requires adjustments to the training strategy. These challenges could limit the usability of AlgoFormer in scenarios where algorithmic structures are complicated to model.

Third, the scalability of AlgoFormer to large-scale applications. The scaling behavior of AlgoFormer has not been thoroughly investigated in this paper. The tasks studied are primarily hand-generated scientific problems or toy cases, and the language task considered is of medium scale. Similarly, the model sizes used in this study are relatively small compared to state-of-the-art large language models (LLMs). To fully assess the potential of AlgoFormer, future research should explore its scalability to larger, more challenging tasks and datasets, as well as the associated computational and memory requirements when deploying AlgoFormer at scale.

## Acknowledgements

M. Ng's research is partly supported by the National Key Research and Development Program of China under Grant 2024YFE0202900; HKRGC GRF 17201020 and 17300021, HKRGC CRF C7004-21GF, and Joint NSFC and RGC N-HKU769/21.

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

# A    Technical Proofs

In this section, we provide comprehensive proofs for all theorems stated in the main content.

**Notation**. We use boldface capital and lowercase letters to denote matrices and vectors respectively. Non-bold letters represent the elements of matrices or vectors, or scalars. For example, $A_{i,j}$ denotes the $(i,j)$-th element of the matrix $\boldsymbol{A}$. We use $\|\cdot\|_2$ to denote the 2-norm (or the maximal singular value) of a matrix.

## A.1    Proof for Theorem 3.1

**Positional embedding**. The role of positional embedding is pivotal in the performance of transformers. Several studies have investigated its impact on natural language processing tasks, as referenced in (Kazemnejad et al., 2023; Ontanon et al., 2022; Press et al., 2022). In our theoretical construction, we deviate from empirical settings by using quasi-orthogonal vectors as positional embedding in each token vector. This choice, also employed by Li et al. (2023); Giannou et al. (2023), is made for theoretical convenience.

**Lemma A.1** (Quasi-orthogonal vectors). *For any fixed $\epsilon > 0$, there exists a set of vectors $\{\boldsymbol{p}_1, \boldsymbol{p}_2, \cdots, \boldsymbol{p}_N\}$ of dimension $\mathcal{O}\left(\log N\right)$ such that $\boldsymbol{p}_i^\top \boldsymbol{p}_i > \boldsymbol{p}_i^\top \boldsymbol{p}_j + \epsilon$ for all $i \neq j$.*

Before going through the details, the following lemma is crucial for understanding transformers as algorithm learners.

**Lemma A.2.** *A one-layer two-head transformer exhibits the capability to implement a single step of gradient descent in multivariate regression.*

*Proof.* Let us consider the input prompt with positional embedding as follows:

$$\boldsymbol{P} := \begin{bmatrix} \boldsymbol{x}_1 & \boldsymbol{0} & \cdots & \boldsymbol{x}_N & \boldsymbol{0} & \boldsymbol{x}_{\text{test}} & \boldsymbol{0} \\ \boldsymbol{0} & \boldsymbol{y}_1 & \cdots & \boldsymbol{0} & \boldsymbol{y}_N & \boldsymbol{0} & \boldsymbol{0} \\ \frac{1}{N}\boldsymbol{x}_1 & \boldsymbol{0} & \cdots & \frac{1}{N}\boldsymbol{x}_N & \boldsymbol{0} & \boldsymbol{0} & \boldsymbol{0} \\ \boldsymbol{A}_k\boldsymbol{x}_1 - \boldsymbol{y}_1 & \boldsymbol{0} & \cdots & \boldsymbol{A}_k\boldsymbol{x}_N - \boldsymbol{y}_N & \boldsymbol{0} & \boldsymbol{A}_k\boldsymbol{x}_{\text{test}} & \boldsymbol{0} \\ 1 & 0 & \cdots & 1 & 0 & 1 & 0 \\ 0 & 1 & \cdots & 0 & 1 & 0 & 1 \\ 0 & 0 & \cdots & 0 & 0 & 1 & 0 \\ 0 & 0 & \cdots & 0 & 0 & 0 & 1 \end{bmatrix},$$

where the 0-1 indicators are used to identify features, labels, and the test data, respectively. We denote the loss function for the multi-variate regression given samples $\{\boldsymbol{x}_1, \boldsymbol{y}_1, \cdots, \boldsymbol{x}_N, \boldsymbol{y}_N\}$ as

$$\mathcal{L}\left(\boldsymbol{A}\right) = \frac{1}{2N}\sum_{j=1}^{N}\|\boldsymbol{A}\boldsymbol{x}_j - \boldsymbol{y}_j\|_2^2,$$

then

$$\frac{\partial \mathcal{L}}{\partial \boldsymbol{A}}\left(\boldsymbol{A}_k\right) = \frac{1}{N}\sum_{j=1}^{N}\left(\boldsymbol{A}_k\boldsymbol{x}_j - \boldsymbol{y}_j\right)\boldsymbol{x}_j^\top.$$

Now, let us define

$$\boldsymbol{W}_Q\boldsymbol{P} = \begin{bmatrix} c\boldsymbol{x}_1 & \boldsymbol{0} & \cdots & c\boldsymbol{x}_N & \boldsymbol{0} & c\boldsymbol{x}_{\text{test}} & \boldsymbol{0} \\ 1 & 1 & \cdots & 1 & 1 & 1 & 1 \end{bmatrix},$$

$$\boldsymbol{W}_K\boldsymbol{P} = \begin{bmatrix} \frac{1}{N}\boldsymbol{x}_1 & \boldsymbol{0} & \cdots & \frac{1}{N}\boldsymbol{x}_N & \boldsymbol{0} & \boldsymbol{0} & \boldsymbol{0} \\ 0 & 0 & \cdots & 0 & 0 & 0 & C \end{bmatrix},$$

and

$$\boldsymbol{W}_V\boldsymbol{P} = e^C \begin{bmatrix} \boldsymbol{A}_k\boldsymbol{x}_1 - \boldsymbol{y}_1 & \boldsymbol{0} & \cdots & \boldsymbol{A}_k\boldsymbol{x}_N - \boldsymbol{y}_N & \boldsymbol{0} & \boldsymbol{A}_k\boldsymbol{x}_{\text{test}} & \boldsymbol{0} \end{bmatrix},$$

for some scalers $C, c > 0$. Here, we denote

$$
\boldsymbol{Z} := \boldsymbol{P}^\top \boldsymbol{W}_K^\top \boldsymbol{W}_Q \boldsymbol{P} =
\begin{bmatrix}
\frac{c}{N}\boldsymbol{x}_1^\top \boldsymbol{x}_1 & 0 & \cdots & \frac{c}{N}\boldsymbol{x}_1^\top \boldsymbol{x}_N & 0 & \frac{c}{N}\boldsymbol{x}_1^\top \boldsymbol{x}_{\text{test}} & 0 \\
\vdots & \vdots & \ddots & \vdots & \vdots & \vdots & \vdots \\
\frac{c}{N}\boldsymbol{x}_N^\top \boldsymbol{x}_1 & 0 & \cdots & \frac{c}{N}\boldsymbol{x}_N^\top \boldsymbol{x}_N & 0 & \frac{c}{N}\boldsymbol{x}_N^\top \boldsymbol{x}_{\text{test}} & 0 \\
0 & 0 & \cdots & 0 & 0 & 0 & 0 \\
0 & 0 & \cdots & 0 & 0 & 0 & 0 \\
C & C & \cdots & C & C & C & C
\end{bmatrix},
$$

then

$$
e^C \cdot \mathrm{softmax}\left(Z_{2i-1,2j-1}\right) \approx 1 + \frac{c}{N}\boldsymbol{x}_i^\top \boldsymbol{x}_j,
$$

where the two sides of the "$\approx$" can be arbitrarily close if $C > 0$ is sufficiently large and $c > 0$ is sufficiently small. The constant here can be canceled by introducing another head. Therefore, the output of the attention layer is

$$
\sum_{i=1}^{2} \boldsymbol{W}_V^{(i)} \boldsymbol{P} \cdot \mathrm{softmax}\left(\boldsymbol{P}^\top \boldsymbol{W}_K^{(i)\top} \boldsymbol{W}_Q^{(i)} \boldsymbol{P}\right) \approx c \left[ \begin{array}{ccccccc} \frac{\partial \mathcal{L}}{\partial \boldsymbol{A}}(\boldsymbol{A}_k)\boldsymbol{x}_1 & \boldsymbol{0} & \cdots & \frac{\partial \mathcal{L}}{\partial \boldsymbol{A}}(\boldsymbol{A}_k)\boldsymbol{x}_N & \boldsymbol{0} & \frac{\partial \mathcal{L}}{\partial \boldsymbol{A}}(\boldsymbol{A}_k)\boldsymbol{x}_{\text{test}} & \boldsymbol{0} \end{array} \right].
$$

The transformer layer's output, after passing through the feed-forward neural network, is expressed as:

$$
\begin{bmatrix}
\boldsymbol{x}_1 & \boldsymbol{0} & \cdots & \boldsymbol{x}_N & \boldsymbol{0} & \boldsymbol{x}_{\text{test}} & \boldsymbol{0} \\
\boldsymbol{0} & \boldsymbol{y}_1 & \cdots & \boldsymbol{0} & \boldsymbol{y}_N & \boldsymbol{0} & \boldsymbol{0} \\
\boldsymbol{A}_{k+1}\boldsymbol{x}_1 - \boldsymbol{y}_1 & \boldsymbol{0} & \cdots & \boldsymbol{A}_{k+1}\boldsymbol{x}_N - \boldsymbol{y}_N & \boldsymbol{0} & \boldsymbol{A}_{k+1}\boldsymbol{x}_{\text{test}} & \boldsymbol{0} \\
1 & 0 & \cdots & 1 & 0 & 1 & 0 \\
0 & 1 & \cdots & 0 & 1 & 0 & 1 \\
0 & 0 & \cdots & 0 & 0 & 1 & 0 \\
0 & 0 & \cdots & 0 & 0 & 0 & 1
\end{bmatrix}.
$$

This signifies the completion of one step of gradient descent with $\boldsymbol{A}_{k+1} = \boldsymbol{A}_k - \eta \frac{\partial \mathcal{L}}{\partial \boldsymbol{A}}(\boldsymbol{A}_k)$ and a positive step size $\eta > 0$. $\qquad \square$

**Proof for Theorem 3.1**. We start by showing that $L$-layer transformer can represent $L$-layer MLPs. It is observed that the identity operation (i.e., $\mathrm{Attn}\,(\boldsymbol{X}) = \boldsymbol{X}$) can be achieved by setting $\boldsymbol{W}_V = \boldsymbol{0}$ due to the residual connection in the attention layer. Each feed-forward neural network in a transformer layer can represent a one-layer MLP. Consequently, the representation function $\Phi^*(\cdot)$ can be realized by $L$-layer transformers. At the output layer of the $L$-th layer transformer, let $\boldsymbol{A}_0$ be an initial guess for the weight. The current output token vectors are then given by:

$$
\boldsymbol{P} :=
\begin{bmatrix}
\Phi^*(\boldsymbol{x}_1) & \boldsymbol{0} & \cdots & \Phi^*(\boldsymbol{x}_N) & \boldsymbol{0} & \Phi^*(\boldsymbol{x}_{\text{test}}) & \boldsymbol{0} \\
\boldsymbol{0} & \boldsymbol{y}_1 & \cdots & \boldsymbol{0} & \boldsymbol{y}_N & \boldsymbol{0} & \boldsymbol{0} \\
\boldsymbol{A}_0\Phi^*(\boldsymbol{x}_1) & \boldsymbol{0} & \cdots & \boldsymbol{A}_0\Phi^*(\boldsymbol{x}_N) & \boldsymbol{0} & \boldsymbol{A}_{k+1}\Phi^*(\boldsymbol{x}_{\text{test}}) & \boldsymbol{0} \\
\boldsymbol{p}_1 & \boldsymbol{p}_1 & \cdots & \boldsymbol{p}_N & \boldsymbol{p}_N & \boldsymbol{p}_{N+1} & \boldsymbol{p}_{N+1} \\
\frac{1}{N} & \frac{1}{N} & \cdots & \frac{1}{N} & \frac{1}{N} & 0 & 0 \\
1 & 0 & \cdots & 1 & 0 & 1 & 0 \\
0 & 1 & \cdots & 0 & 1 & 0 & 1 \\
0 & 0 & \cdots & 0 & 0 & 1 & 0 \\
0 & 0 & \cdots & 0 & 0 & 0 & 1
\end{bmatrix}.
$$

Here, the set of quasi-orthogonal vectors $\{\boldsymbol{p}_1, \cdots, \boldsymbol{p}_{N+1}\}$ is generated, according to Lemma A.1. The next transformer layer is designed to facilitate the exchange of information between neighboring tokens. Let

$$
\boldsymbol{W}_K \boldsymbol{P} = \boldsymbol{W}_Q \boldsymbol{P} = \left[ \begin{array}{cccccc} \boldsymbol{p}_1 & \boldsymbol{p}_1 & \cdots & \boldsymbol{p}_N & \boldsymbol{p}_N & \boldsymbol{p}_{N+1} & \boldsymbol{p}_{N+1} \end{array} \right]
$$

and

$$
\boldsymbol{W}_V \boldsymbol{P} = \left[ \begin{array}{ccccccc} \boldsymbol{0} & \boldsymbol{y}_1 & \cdots & \boldsymbol{0} & \boldsymbol{y}_N & \boldsymbol{0} & \boldsymbol{0} \end{array} \right],
$$

then

$$\boldsymbol{W}_V \boldsymbol{P} \cdot \mathrm{softmax}\left(\boldsymbol{P}^\top \boldsymbol{W}_K^\top \boldsymbol{W}_Q \boldsymbol{P}\right) \approx \begin{bmatrix} \frac{1}{2}\boldsymbol{y}_1 & \frac{1}{2}\boldsymbol{y}_1 & \cdots & \frac{1}{2}\boldsymbol{y}_N & \frac{1}{2}\boldsymbol{y}_N & \boldsymbol{0} & \boldsymbol{0} \end{bmatrix},$$

where the two sides of the "$\approx$" can be arbitrarily close if the temperature of the softmax function is sufficiently large, due to the nearly orthogonality of positional embedding vectors. It's important to note that the feed-forward neural network is capable of approximating nonlinear functions, such as multiplication. Here, we construct a shallow neural network that calculates the multiplication between the first $d$ elements and the value $\frac{1}{N}$ in each token. Passing through the feed-forward neural network together with the indicators, we obtain the final output of the first $(L+1)$-layer transformer $\mathrm{TF}_{\mathrm{pre}}$:

$$\begin{bmatrix} \Phi^*\left(\boldsymbol{x}_1\right) & \boldsymbol{0} & \cdots & \Phi^*\left(\boldsymbol{x}_N\right) & \boldsymbol{0} & \Phi^*\left(\boldsymbol{x}_{\mathrm{test}}\right) & \boldsymbol{0} \\ \boldsymbol{0} & \boldsymbol{y}_1 & \cdots & \boldsymbol{0} & \boldsymbol{y}_N & \boldsymbol{0} & \boldsymbol{0} \\ \frac{1}{N}\Phi^*\left(\boldsymbol{x}_1\right) & \boldsymbol{0} & \cdots & \frac{1}{N}\Phi^*\left(\boldsymbol{x}_N\right) & \boldsymbol{0} & \boldsymbol{0} & \boldsymbol{0} \\ \boldsymbol{A}_0\Phi^*\left(\boldsymbol{x}_1\right) - \boldsymbol{y}_1 & \boldsymbol{0} & \cdots & \boldsymbol{A}_0\Phi^*\left(\boldsymbol{x}_N\right) - \boldsymbol{y}_N & \boldsymbol{0} & \boldsymbol{A}_0\Phi^*\left(\boldsymbol{x}_{\mathrm{test}}\right) & \boldsymbol{0} \\ \boldsymbol{p}_1 & \boldsymbol{p}_1 & \cdots & \boldsymbol{p}_N & \boldsymbol{p}_N & \boldsymbol{p}_{N+1} & \boldsymbol{p}_{N+1} \\ 1 & 0 & \cdots & 1 & 0 & 1 & 0 \\ 0 & 1 & \cdots & 0 & 1 & 0 & 1 \\ 0 & 0 & \cdots & 0 & 0 & 1 & 0 \\ 0 & 0 & \cdots & 0 & 0 & 0 & 1 \end{bmatrix}.$$

According to the construction outlined in Lemma A.2, there exists a one-layer, two-head transformer $\mathrm{TF}_{\mathrm{loop}}$, independent of the exact values of input data samples (tokens), that can implement gradient descent for finding the optimal weight $\boldsymbol{A}^*$ in the context of multivariate regression. The optimization aims to minimize the following empirical loss:

$$\min_{\boldsymbol{A}} \frac{1}{2N} \sum_{i=1}^{N} \left\| \boldsymbol{A}\Phi^*\left(\boldsymbol{x}_i\right) - \boldsymbol{y}_i \right\|_2^2.$$

After $k$-steps of looped transformer $\mathrm{TF}_{\mathrm{loop}}$, which corresponds to applying $k$ steps of gradient descent, the resulting token vectors follows

$$\begin{bmatrix} \Phi^*\left(\boldsymbol{x}_1\right) & \boldsymbol{0} & \cdots & \Phi^*\left(\boldsymbol{x}_N\right) & \boldsymbol{0} & \Phi^*\left(\boldsymbol{x}_{\mathrm{test}}\right) & \boldsymbol{0} \\ \boldsymbol{0} & \boldsymbol{y}_1 & \cdots & \boldsymbol{0} & \boldsymbol{y}_N & \boldsymbol{0} & \boldsymbol{0} \\ \frac{1}{N}\Phi^*\left(\boldsymbol{x}_1\right) & \boldsymbol{0} & \cdots & \frac{1}{N}\Phi^*\left(\boldsymbol{x}_N\right) & \boldsymbol{0} & \boldsymbol{0} & \boldsymbol{0} \\ \boldsymbol{A}_k\Phi^*\left(\boldsymbol{x}_1\right) - \boldsymbol{y}_1 & \boldsymbol{0} & \cdots & \boldsymbol{A}_k\Phi^*\left(\boldsymbol{x}_N\right) - \boldsymbol{y}_N & \boldsymbol{0} & \boldsymbol{A}_k\Phi^*\left(\boldsymbol{x}_{\mathrm{test}}\right) & \boldsymbol{0} \\ \boldsymbol{p}_1 & \boldsymbol{p}_1 & \cdots & \boldsymbol{p}_N & \boldsymbol{p}_N & \boldsymbol{p}_{N+1} & \boldsymbol{p}_{N+1} \\ 1 & 0 & \cdots & 1 & 0 & 1 & 0 \\ 0 & 1 & \cdots & 0 & 1 & 0 & 1 \\ 0 & 0 & \cdots & 0 & 0 & 1 & 0 \\ 0 & 0 & \cdots & 0 & 0 & 0 & 1 \end{bmatrix}.$$

These token vectors are then ready for processing by the output transformer layer $\mathrm{TF}_{\mathrm{post}}$. The post-transformer is designed to facilitate communication between the last two tokens and position the desired result $\boldsymbol{A}_k\Phi^*\left(\boldsymbol{x}_{\mathrm{test}}\right)$ in the appropriate position. We can similarly set

$$\boldsymbol{W}_K \boldsymbol{P} = \boldsymbol{W}_Q \boldsymbol{P} = \begin{bmatrix} \boldsymbol{p}_1 & \boldsymbol{p}_1 & \cdots & \boldsymbol{p}_N & \boldsymbol{p}_N & \boldsymbol{p}_{N+1} & \boldsymbol{p}_{N+1} \end{bmatrix},$$

$$\boldsymbol{W}_V \boldsymbol{P} = \begin{bmatrix} \boldsymbol{A}_k\Phi^*\left(\boldsymbol{x}_1\right) - \boldsymbol{y}_1 & \boldsymbol{0} & \cdots & \boldsymbol{A}_k\Phi^*\left(\boldsymbol{x}_N\right) - \boldsymbol{y}_N & \boldsymbol{0} & \boldsymbol{A}_k\Phi^*\left(\boldsymbol{x}_{\mathrm{test}}\right) & \boldsymbol{0} \end{bmatrix},$$

and pass it through the feed-forward neural network. This results in the final output:

$$\begin{bmatrix} \Phi^*\left(\boldsymbol{x}_1\right) & \boldsymbol{0} & \cdots & \Phi^*\left(\boldsymbol{x}_N\right) & \boldsymbol{0} & \Phi^*\left(\boldsymbol{x}_{\mathrm{test}}\right) & \boldsymbol{0} \\ \boldsymbol{0} & \boldsymbol{y}_1 & \cdots & \boldsymbol{0} & \boldsymbol{y}_N & \boldsymbol{0} & \boldsymbol{A}_k\Phi^*\left(\boldsymbol{x}_{\mathrm{test}}\right) \end{bmatrix},$$

which completes the proof.

## A.2 Proof for Theorem 3.2

The following lemma highlights the intrinsic "copying" capability of transformers, a pivotal feature for autoregressive models, especially in the context of time series analysis.

**Lemma A.3.** *A one-layer transformer with q heads possesses the ability to effectively copy information from the previous q tokens to the present token.*

*Proof.* We construct the input prompt with positional embedding as follows:

$$
\begin{bmatrix}
\boldsymbol{x}_0 & \boldsymbol{x}_1 & \boldsymbol{x}_2 & \cdots & \boldsymbol{x}_N \\
\boldsymbol{p}_0 & \boldsymbol{p}_1 & \boldsymbol{p}_2 & \cdots & \boldsymbol{p}_N \\
\boldsymbol{p}_{-1} & \boldsymbol{p}_0 & \boldsymbol{p}_1 & \cdots & \boldsymbol{p}_{N-1} \\
\vdots & \vdots & \vdots & \ddots & \vdots \\
\boldsymbol{p}_{-q} & \boldsymbol{p}_{1-q} & \boldsymbol{p}_{2-q} & \cdots & \boldsymbol{p}_{N-q} \\
0 & 1 & 2 & \cdots & N
\end{bmatrix}.
$$

The i-th head aims to connect and communicate the current token with the previous $i$-th token. Specifically, we let

$$
\boldsymbol{W}_K^{(i)} \boldsymbol{P} = \begin{bmatrix} \boldsymbol{p}_0 & \boldsymbol{p}_1 & \boldsymbol{p}_2 & \cdots & \boldsymbol{p}_N \end{bmatrix},
$$

$$
\boldsymbol{W}_Q^{(i)} \boldsymbol{P} = \begin{bmatrix} \boldsymbol{p}_{-i} & \boldsymbol{p}_{1-i} & \boldsymbol{p}_{2-i} & \cdots & \boldsymbol{p}_{N-i} \end{bmatrix},
$$

and

$$
\boldsymbol{W}_V^{(i)} \boldsymbol{P} = \begin{bmatrix} \boldsymbol{x}_0 & \boldsymbol{x}_1 & \boldsymbol{x}_2 & \cdots & \boldsymbol{x}_N \end{bmatrix},
$$

we have

$$
\boldsymbol{W}_V^{(i)} \boldsymbol{P} \cdot \mathrm{softmax}\left( \boldsymbol{P}^\top \boldsymbol{W}_K^{(i)\top} \boldsymbol{W}_Q^{(i)} \boldsymbol{P} \right) \approx \begin{bmatrix} * & \boldsymbol{x}_{1-i} & \boldsymbol{x}_{2-i} & \cdots & \boldsymbol{x}_{N-i} \end{bmatrix}.
$$

Here, we use "*" to mask some unimportant token values. Therefore, the $q$-head attention layer outputs

$$
\begin{bmatrix}
\boldsymbol{x}_0 & \boldsymbol{x}_1 & \boldsymbol{x}_2 & \cdots & \boldsymbol{x}_N \\
* & \boldsymbol{x}_0 & \boldsymbol{x}_1 & \cdots & \boldsymbol{x}_{N-1} \\
* & * & \boldsymbol{x}_0 & \cdots & \boldsymbol{x}_{N-2} \\
\vdots & \vdots & \vdots & \ddots & \vdots \\
* & * & * & \cdots & \boldsymbol{x}_{N-q} \\
0 & 1 & 2 & \cdots & N
\end{bmatrix}.
$$

Here, the samples are only supported on $\{\boldsymbol{x}_t\}$ with $0 \le t \le N$. It is common to alternatively define $\boldsymbol{x}_t = \boldsymbol{0}$ for $t < 0$. Passing through the feed-forward neural network, together with the indicators at the last row, we can filter out the undefined elements, i.e.,

$$
\begin{bmatrix}
\boldsymbol{x}_0 & \boldsymbol{x}_1 & \boldsymbol{x}_2 & \cdots & \boldsymbol{x}_N \\
\boldsymbol{0} & \boldsymbol{x}_0 & \boldsymbol{x}_1 & \cdots & \boldsymbol{x}_{N-1} \\
\boldsymbol{0} & \boldsymbol{0} & \boldsymbol{x}_0 & \cdots & \boldsymbol{x}_{N-2} \\
\vdots & \vdots & \vdots & \ddots & \vdots \\
\boldsymbol{0} & \boldsymbol{0} & \boldsymbol{0} & \cdots & \boldsymbol{x}_{N-q} \\
0 & 1 & 2 & \cdots & N
\end{bmatrix}.
$$

$\square$

**Proof for Theorem 3.2** According to Lemma A.3 and Theorem 3.1, the $(L+2)$-layer pre-transformer $\mathrm{TF}_{\mathrm{pre}}$ is able to do copying and transformation of the representation function. The output after the preprocessing is given by

$$
\begin{bmatrix}
\boldsymbol{x}_0 & \boldsymbol{x}_1 & \boldsymbol{x}_2 & \cdots & \boldsymbol{x}_N \\
\Phi^*\left(\boldsymbol{x}_{1-q:0}\right) & \Phi^*\left(\boldsymbol{x}_{2-q:1}\right) & \Phi^*\left(\boldsymbol{x}_{3-q:2}\right) & \cdots & \Phi^*\left(\boldsymbol{x}_{N+1-q:N}\right)
\end{bmatrix},
$$

where we denote $\boldsymbol{x}_{i:j}$ as the concatenation $[\boldsymbol{x}_i, \boldsymbol{x}_{i+1}, \cdots, \boldsymbol{x}_j]$ for notational simplicity. Similar to the construction in Lemma A.2, a one-layer two-head transformer $\mathrm{TF}_{\mathrm{loop}}$ is capable of implementing gradient descent on the multivariate regression. Finally, the post-transformer $\mathrm{TF}_{\mathrm{post}}$ moves the desired result to the output.

## A.3 Proof for Theorem 3.3

According to Lemma 5 in (Li et al., 2023), a seven-layer, two-head pre-transformer $\mathrm{TF}_{\mathrm{pre}}$ is introduced for preprocessing the input CoT sequence. This pre-transformer performs filtering, transforming the input sequence into the structured form given by Equation 7.

$$\begin{bmatrix} \mathbf{0} & \cdots & \boldsymbol{s}_1^{\ell-1} & \mathbf{0} & \cdots & \boldsymbol{s}_2^{\ell-1} & \mathbf{0} & \cdots & \mathbf{0} & \mathbf{0} & \cdots & \hat{\boldsymbol{s}}^{\ell-1} \\ \mathbf{0} & \cdots & \mathbf{0} & \boldsymbol{s}_1^{\ell} & \cdots & \mathbf{0} & \boldsymbol{s}_2^{\ell} & \cdots & \mathbf{0} & \mathbf{0} & \cdots & \mathbf{0} \\ 0 & \cdots & 1 & 0 & \cdots & 1 & 0 & \cdots & 0 & 0 & \cdots & 1 \\ 0 & \cdots & 0 & 1 & \cdots & 0 & 1 & \cdots & 0 & 0 & \cdots & 0 \\ 0 & \cdots & 0 & 0 & \cdots & 0 & 0 & \cdots & 0 & 0 & \cdots & 1 \end{bmatrix}. \tag{7}$$

Specifically, it identifies the positional index $\ell-1$ of the last token $\hat{\boldsymbol{s}}^{\ell-1}$, retains only $\boldsymbol{s}_i^{\ell-1}$ and $\boldsymbol{s}_i^{\ell}$ for $1 \leq i \leq N$, and filters out all other irrelevant tokens. In this context, the representation function $\Phi^*(\cdot)$ corresponds to $L$-layer leaky ReLU MLPs. Notably, the transformation $\boldsymbol{s}i^{\ell} = \sigma\left(\boldsymbol{W}^{\ell}\boldsymbol{s}i^{\ell-1}\right)$ is expressed, where $\boldsymbol{W}^{\ell}$ denotes the weight matrix at the $\ell$-th layer, and $\sigma(\cdot)$ represents the leaky ReLU activation function. Given the reversibility and piecewise linearity of the leaky ReLU activation, we can assume, without loss of generality, that $\boldsymbol{s}_i^{\ell} = \boldsymbol{W}^{\ell}\boldsymbol{s}_i^{\ell-1}$ in Equation 7. Consequently, the problem is reduced to a multi-variate regression, and a one-layer two-head transformer $\mathrm{TF}_{\mathrm{loop}}$ is demonstrated to effectively implement gradient descent for determining the weight matrix $\boldsymbol{W}^{\ell}$, as shown in Lemma A.2. Subsequently, the post-transformer $\mathrm{TF}_{\mathrm{post}}$ produces the desired result $\sigma\left(\boldsymbol{W}^{\ell}\hat{\boldsymbol{s}}i^{\ell-1}\right)$.

## A.4 Proof for Theorem 4.1

In this section, we first show that the one-layer two-head transformer can implement a single step of Newton's method in Equation 5, with the special form of input token vectors. Then, we introduce the pre-transformer, designed to convert general input tokens into the prescribed format conducive to the transformer's operation. Finally, the post-transformer facilitates the extraction of the desired results through additional computations, given that the output from the looped-transformer corresponds to an intermediate product.

**Lemma A.4.** *A transformer with one layer and two heads is capable of implementing one step of Newton's method in the linear regression problem in Equation 4.*

*Proof.* Let us consider the input prompt with positional embedding as follows:

$$\boldsymbol{P} := \begin{bmatrix} \boldsymbol{x}_1 & \mathbf{0} & \cdots & \boldsymbol{x}_N & \mathbf{0} & \boldsymbol{x}_{\text{test}} & \mathbf{0} \\ 0 & y_1 & \cdots & 0 & y_N & 0 & 0 \\ \boldsymbol{M}_k\boldsymbol{x}_1 & \mathbf{0} & \cdots & \boldsymbol{M}_k\boldsymbol{x}_N & \mathbf{0} & \mathbf{0} & \mathbf{0} \\ 1 & 0 & \cdots & 1 & 0 & 0 & 0 \\ 0 & 1 & \cdots & 0 & 1 & 0 & 0 \\ 0 & 0 & \cdots & 0 & 0 & 1 & 0 \\ 0 & 0 & \cdots & 0 & 0 & 0 & 1 \end{bmatrix}.$$

Let

$$\boldsymbol{W}_Q\boldsymbol{P} = \begin{bmatrix} \boldsymbol{M}_k\boldsymbol{x}_1 & \mathbf{0} & \cdots & \boldsymbol{M}_k\boldsymbol{x}_N & \mathbf{0} & \mathbf{0} & \mathbf{0} \\ 1 & 1 & \cdots & 1 & 1 & 1 & 1 \end{bmatrix},$$

$$\boldsymbol{W}_K\boldsymbol{P} = \begin{bmatrix} c\boldsymbol{x}_1 & \mathbf{0} & \cdots & c\boldsymbol{x}_N & \mathbf{0} & c\boldsymbol{x}_{\text{test}} & \mathbf{0} \\ 0 & 0 & \cdots & 0 & 0 & 0 & C \end{bmatrix},$$

and

$$\boldsymbol{W}_V\boldsymbol{P} = e^C \begin{bmatrix} \boldsymbol{M}_k\boldsymbol{x}_1 & \mathbf{0} & \cdots & \boldsymbol{M}_k\boldsymbol{x}_N & \mathbf{0} & \mathbf{0} & \mathbf{0} \end{bmatrix}.$$

Similarly, denote $\boldsymbol{Z} := \boldsymbol{P}^{\top}\boldsymbol{W}_K^{\top}\boldsymbol{W}_Q\boldsymbol{P}$, we can establish that

$$e^C \mathrm{softmax}\left(Z_{2i-1,2j-1}\right) \approx 1 + c\boldsymbol{x}_i^{\top}\boldsymbol{M}_k\boldsymbol{x}_j. \tag{8}$$

To nullify the constant term, an additional attention head can be incorporated. Therefore, the output takes the form:

$$\boldsymbol{W}_V \boldsymbol{P}\mathrm{softmax}\left(\boldsymbol{P}^\top \boldsymbol{W}_K^\top \boldsymbol{W}_Q \boldsymbol{P}\right) \approx \left[\begin{array}{cccccc} c\boldsymbol{M}_k \boldsymbol{S}\boldsymbol{M}_k \boldsymbol{x}_1 & * & \cdots & c\boldsymbol{M}_k \boldsymbol{S}\boldsymbol{M}_k \boldsymbol{x}_N & * & * \end{array}\right].$$

Here, we use "*" to mask some unimportant token values. Upon passing through the feed-forward neural network with indicator $\left[\begin{array}{ccccccc} 1 & 0 & \cdots & 1 & 0 & 0 & 0 \end{array}\right]$ and weight $1/c$, the resulting output is

$$\begin{bmatrix} \boldsymbol{x}_1 & \boldsymbol{0} & \cdots & \boldsymbol{x}_N & \boldsymbol{0} & \boldsymbol{x}_{\mathrm{test}} & \boldsymbol{0} \\ 0 & y_1 & \cdots & 0 & y_N & 0 & 0 \\ \boldsymbol{M}_{k+1}\boldsymbol{x}_1 & \boldsymbol{0} & \cdots & \boldsymbol{M}_{k+1}\boldsymbol{x}_N & \boldsymbol{0} & \boldsymbol{0} & \boldsymbol{0} \\ 1 & 0 & \cdots & 1 & 0 & 0 & 0 \\ 0 & 1 & \cdots & 0 & 1 & 0 & 0 \\ 0 & 0 & \cdots & 0 & 0 & 1 & 0 \\ 0 & 0 & \cdots & 0 & 0 & 0 & 1 \end{bmatrix},$$

where $\boldsymbol{M}_{k+1} = 2\boldsymbol{M}_k - \boldsymbol{M}_k \boldsymbol{S}\boldsymbol{M}_k$. □

**Proof for Theorem 4.1**. For $\mathrm{TF}_{\mathrm{pre}}$, we adopt the following configurations:

$$\boldsymbol{W}_Q \boldsymbol{P} = \left[\begin{array}{ccccccc} \boldsymbol{x}_1 & \boldsymbol{0} & \cdots & \boldsymbol{x}_N & \boldsymbol{0} & \boldsymbol{0} & \boldsymbol{0} \\ 1 & 1 & \cdots & 1 & 1 & 1 & 1 \end{array}\right],$$

$$\boldsymbol{W}_K \boldsymbol{P} = \left[\begin{array}{ccccccc} c\boldsymbol{x}_1 & \boldsymbol{0} & \cdots & c\boldsymbol{x}_N & \boldsymbol{0} & c\boldsymbol{x}_{\mathrm{test}} & \boldsymbol{0} \\ 0 & 0 & \cdots & 0 & 0 & 0 & C \end{array}\right],$$

and

$$\boldsymbol{W}_V \boldsymbol{P} = e^C \left[\begin{array}{ccccccc} \boldsymbol{x}_1 & \boldsymbol{0} & \cdots & \boldsymbol{x}_N & \boldsymbol{0} & \boldsymbol{0} & \boldsymbol{0} \end{array}\right].$$

Denote $\boldsymbol{Z} := \boldsymbol{P}^\top \boldsymbol{W}_K^\top \boldsymbol{W}_Q \boldsymbol{P}$, we can show that

$$e^C \mathrm{softmax}\left(Z_{2i-1,2j-1}\right) \approx 1 + c\boldsymbol{x}_i^\top \boldsymbol{x}_j.$$

We may include another attention head to remove the constant. Therefore, the output is formulated as

$$\boldsymbol{W}_V \boldsymbol{P}\mathrm{softmax}\left(\boldsymbol{P}^\top \boldsymbol{W}_K^\top \boldsymbol{W}_Q \boldsymbol{P}\right) \approx \left[\begin{array}{cccccc} c\boldsymbol{S}\boldsymbol{x}_1 & * & \cdots & c\boldsymbol{S}\boldsymbol{x}_N & * & * \end{array}\right].$$

After passing through the feed-forward neural network with indicators $\left[\begin{array}{ccccccc} 1 & 0 & \cdots & 1 & 0 & 0 & 0 \end{array}\right]$ and weight $\alpha/c$, the resulting output becomes:

$$\begin{bmatrix} \boldsymbol{x}_1 & \boldsymbol{0} & \cdots & \boldsymbol{x}_N & \boldsymbol{0} & \boldsymbol{x}_{\mathrm{test}} & \boldsymbol{0} \\ 0 & y_1 & \cdots & 0 & y_N & 0 & 0 \\ \boldsymbol{M}_0\boldsymbol{x}_1 & \boldsymbol{0} & \cdots & \boldsymbol{M}_0\boldsymbol{x}_N & \boldsymbol{0} & \boldsymbol{0} & \boldsymbol{0} \\ 1 & 0 & \cdots & 1 & 0 & 0 & 0 \\ 0 & 1 & \cdots & 0 & 1 & 0 & 0 \\ 0 & 0 & \cdots & 0 & 0 & 1 & 0 \\ 0 & 0 & \cdots & 0 & 0 & 0 & 1 \end{bmatrix},$$

where $\boldsymbol{M}_0 = \alpha\boldsymbol{S}$.

As illustrated in Lemma A.4, after $T$ iterations of the looped transformer $\mathrm{TF}_{\mathrm{loop}}$, it produces the following output:

$$\begin{bmatrix} \boldsymbol{x}_1 & \boldsymbol{0} & \cdots & \boldsymbol{x}_N & \boldsymbol{0} & \boldsymbol{x}_{\mathrm{test}} & \boldsymbol{0} \\ 0 & y_1 & \cdots & 0 & y_N & 0 & 0 \\ \boldsymbol{M}_T\boldsymbol{x}_1 & \boldsymbol{0} & \cdots & \boldsymbol{M}_T\boldsymbol{x}_N & \boldsymbol{0} & \boldsymbol{0} & \boldsymbol{0} \\ 1 & 0 & \cdots & 1 & 0 & 0 & 0 \\ 0 & 1 & \cdots & 0 & 1 & 0 & 0 \\ 0 & 0 & \cdots & 0 & 0 & 1 & 0 \\ 0 & 0 & \cdots & 0 & 0 & 0 & 1 \end{bmatrix}.$$

In the post-transformer, additional positional embeddings are introduced to address technical considerations. The input is structured as follows:

$$\begin{bmatrix} \boldsymbol{x}_1 & \boldsymbol{0} & \cdots & \boldsymbol{x}_N & \boldsymbol{0} & \boldsymbol{x}_{\text{test}} & \boldsymbol{0} \\ 0 & y_1 & \cdots & 0 & y_N & 0 & 0 \\ \boldsymbol{M}_T\boldsymbol{x}_1 & \boldsymbol{0} & \cdots & \boldsymbol{M}_T\boldsymbol{x}_N & \boldsymbol{0} & \boldsymbol{0} & \boldsymbol{0} \\ \boldsymbol{p}_1 & \boldsymbol{p}_1 & \cdots & \boldsymbol{p}_N & \boldsymbol{p}_N & \boldsymbol{p}_{N+1} & \boldsymbol{p}_{N+1} \\ 1 & 0 & \cdots & 1 & 0 & 0 & 0 \\ 0 & 1 & \cdots & 0 & 1 & 0 & 0 \\ 0 & 0 & \cdots & 0 & 0 & 1 & 0 \\ 0 & 0 & \cdots & 0 & 0 & 0 & 1 \end{bmatrix},$$

where the positional embedding vectors $\boldsymbol{p}_1, \cdots, \boldsymbol{p}_{N+1}$ are designed to be nearly orthogonal (Lemma A.1). To initiate the weight $\boldsymbol{w}_T^{\text{Newton}}$, we propagate the target label $y$ to adjacent tokens using the following attention mechanism:

$$\boldsymbol{W}_K\boldsymbol{P} = \boldsymbol{W}_Q\boldsymbol{P} = \begin{bmatrix} \boldsymbol{p}_1 & \boldsymbol{p}_1 & \cdots & \boldsymbol{p}_N & \boldsymbol{p}_N & \boldsymbol{p}_{N+1} & \boldsymbol{p}_{N+1} \end{bmatrix}$$

and

$$\boldsymbol{W}_V\boldsymbol{P} = 2 \begin{bmatrix} 0 & y_1 & \cdots & 0 & y_N & 0 & 0 \end{bmatrix}.$$

This operation results in the attention layer producing the following output:

$$\begin{bmatrix} \boldsymbol{x}_1 & \boldsymbol{0} & \cdots & \boldsymbol{x}_N & \boldsymbol{0} & \boldsymbol{x}_{\text{test}} & \boldsymbol{0} \\ y_1 & y_1 & \cdots & y_N & y_N & 0 & 0 \\ \boldsymbol{M}_T\boldsymbol{x}_1 & \boldsymbol{0} & \cdots & \boldsymbol{M}_T\boldsymbol{x}_N & \boldsymbol{0} & \boldsymbol{0} & \boldsymbol{0} \\ \boldsymbol{p}_1 & \boldsymbol{p}_1 & \cdots & \boldsymbol{p}_N & \boldsymbol{p}_N & \boldsymbol{p}_{N+1} & \boldsymbol{p}_{N+1} \\ 1 & 0 & \cdots & 1 & 0 & 0 & 0 \\ 0 & 1 & \cdots & 0 & 1 & 0 & 0 \\ 0 & 0 & \cdots & 0 & 0 & 1 & 0 \\ 0 & 0 & \cdots & 0 & 0 & 0 & 1 \end{bmatrix}.$$

In the next layer, analogous to the construction in Equation 8, we define the following transformations:

$$\boldsymbol{W}_Q\boldsymbol{P} = \begin{bmatrix} c\boldsymbol{x}_1 & \boldsymbol{0} & \cdots & c\boldsymbol{x}_N & \boldsymbol{0} & c\boldsymbol{x}_{\text{test}} & \boldsymbol{0} \\ 1 & 1 & \cdots & 1 & 1 & 1 & 1 \end{bmatrix},$$

$$\boldsymbol{W}_K\boldsymbol{P} = \begin{bmatrix} \boldsymbol{M}_T\boldsymbol{x}_1 & \boldsymbol{0} & \cdots & \boldsymbol{M}_T\boldsymbol{x}_N & \boldsymbol{0} & \boldsymbol{0} & \boldsymbol{0} \\ 0 & 0 & \cdots & 0 & 0 & 0 & C \end{bmatrix},$$

and

$$\boldsymbol{W}_V\boldsymbol{P} = e^C \begin{bmatrix} y_1 & y_1 & \cdots & y_N & y_N & 0 & 0 \end{bmatrix},$$

for some $C, c > 0$. Defining the matrix

$$\boldsymbol{Z} := \boldsymbol{P}^\top\boldsymbol{W}_K^\top\boldsymbol{W}_Q\boldsymbol{P} = \begin{bmatrix} c\boldsymbol{x}_1^\top\boldsymbol{M}_T\boldsymbol{x}_1 & 0 & \cdots & c\boldsymbol{x}_1^\top\boldsymbol{M}_T\boldsymbol{x}_N & 0 & c\boldsymbol{x}_1^\top\boldsymbol{M}_T\boldsymbol{x}_{\text{test}} & 0 \\ \vdots & \vdots & \ddots & \vdots & \vdots & \vdots & \vdots \\ c\boldsymbol{x}_N^\top\boldsymbol{M}_T\boldsymbol{x}_1 & 0 & \cdots & c\boldsymbol{x}_N^\top\boldsymbol{M}_T\boldsymbol{x}_N & 0 & c\boldsymbol{x}_N^\top\boldsymbol{M}_T\boldsymbol{x}_{\text{test}} & 0 \\ 0 & 0 & \cdots & 0 & 0 & 0 & 0 \\ 0 & 0 & \cdots & 0 & 0 & 0 & 0 \\ C & C & \cdots & C & C & C & C \end{bmatrix},$$

we can show that

$$e^C \cdot \text{softmax}\left(Z_{2i-1,2j-1}\right) \approx 1 + c\boldsymbol{x}_i^\top\boldsymbol{M}_T\boldsymbol{x}_j,$$

where the closeness of the two sides of the approximation "$\approx$" can be achieved by selecting $C > 0$ sufficiently large and $c > 0$ sufficiently small. The constant term can be removed by introducing another head. Therefore, the output of the attention layer is expressed as

$$\sum_{i=1}^2 \boldsymbol{W}_V^{(i)}\boldsymbol{X} \cdot \text{softmax}\left(\boldsymbol{X}^\top\boldsymbol{W}_K^{(i)\top}\boldsymbol{W}_Q^{(i)}\boldsymbol{X}\right) \approx c \begin{bmatrix} * & * & \cdots & * & * & \boldsymbol{w}_T^{\text{Newton}\top}\boldsymbol{x}_{\text{test}} & * \end{bmatrix}.$$

Finally, the feed-forward neural network yields the desired prediction $\boldsymbol{w}_T^{\text{Newton}\top}\boldsymbol{x}_{\text{test}}$.

### A.5 Proof for Theorem 4.2

In this section, we extend the realization of gradient descent, as demonstrated in Lemma A.2 for encoder-based transformers, to decoder-based transformers. Although the construction is similar, the key distinction lies in the decoder-based transformer's utilization of previously viewed data for regression, consistent with our intuitive understanding. The following lemma is enough to conclude the proof for Theorem 4.2.

**Lemma A.5.** *The one-layer two-head decoder-based transformer can implement one step of gradient descent in linear regression problems in Equation 6.*

*Proof.* We consider the input prompt with positional embedding as follows:

$$
\begin{bmatrix}
\boldsymbol{x}_1 & \boldsymbol{0} & \cdots & \boldsymbol{x}_{i-1} & \boldsymbol{0} & \boldsymbol{x}_i \\
0 & y_1 & \cdots & 0 & y_{i-1} & 0 \\
\boldsymbol{0} & \boldsymbol{x}_1 & \cdots & \boldsymbol{0} & \boldsymbol{x}_{i-1} & \boldsymbol{0} \\
\boldsymbol{w}_k^1 & \boldsymbol{0} & \cdots & \boldsymbol{w}_k^{i-1} & \boldsymbol{0} & \boldsymbol{w}_k^i \\
0 & 0 & \cdots & \frac{1}{i-2} & 0 & \frac{1}{i-1} \\
1 & 0 & \cdots & 1 & 0 & 1 \\
1 & 0 & \cdots & 0 & 0 & 0 \\
0 & 1 & \cdots & 0 & 0 & 0
\end{bmatrix}.
$$

We construct the attention layer with

$$
\boldsymbol{W}_Q \boldsymbol{P}^i = \begin{bmatrix} \boldsymbol{w}_k^1 & \boldsymbol{0} & \cdots & \boldsymbol{w}_k^{i-1} & \boldsymbol{0} & \boldsymbol{w}_k^i \\ 1 & 1 & \cdots & 1 & 1 & 1 \end{bmatrix},
$$

$$
\boldsymbol{W}_K \boldsymbol{P}^i = \begin{bmatrix} c\boldsymbol{x}_1 & \boldsymbol{0} & \cdots & c\boldsymbol{x}_{i-1} & \boldsymbol{0} & c\boldsymbol{x}_i \\ 0 & C & \cdots & 0 & 0 & 0 \end{bmatrix},
$$

and

$$
\boldsymbol{W}_V \boldsymbol{P}^i = e^C/c \begin{bmatrix} \boldsymbol{x}_1 & \boldsymbol{0} & \cdots & \boldsymbol{x}_{i-1} & \boldsymbol{0} & \boldsymbol{x}_i \end{bmatrix}.
$$

Here, we adopt causal attention, where the attention mechanism can only attend to previous tokens. The output is

$$
\begin{bmatrix} * & * & \cdots & \sum_{j=1}^{i-2} \boldsymbol{w}_k^{i-1\top} \boldsymbol{x}_j \boldsymbol{x}_j & * & \sum_{j=1}^{i-1} \boldsymbol{w}_k^{i\top} \boldsymbol{x}_j \boldsymbol{x}_j \end{bmatrix}.
$$

For the second head, we similarly let

$$
\boldsymbol{W}_Q \boldsymbol{P}^i = \begin{bmatrix} 1 & 0 & \cdots & 1 & 0 & 1 \\ 1 & 0 & \cdots & 1 & 0 & 1 \end{bmatrix},
$$

$$
\boldsymbol{W}_K \boldsymbol{P}^i = \begin{bmatrix} 0 & cy_1 & \cdots & 0 & cy_{i-1} & 0 \\ C & 0 & \cdots & 0 & 0 & 0 \end{bmatrix},
$$

and

$$
\boldsymbol{W}_V \boldsymbol{P}^i = -e^C/c \begin{bmatrix} \boldsymbol{0} & \boldsymbol{x}_1 & \cdots & \boldsymbol{0} & \boldsymbol{x}_{i-1} & \boldsymbol{0} \end{bmatrix}.
$$

Then, we have the output

$$
\begin{bmatrix} * & * & \cdots & -\sum_{j=1}^{i-2} y_j \boldsymbol{x}_j & * & -\sum_{j=1}^{i-1} y_j \boldsymbol{x}_j \end{bmatrix}
$$

The attention layer outputs

$$
\begin{bmatrix}
\boldsymbol{x}_1 & \boldsymbol{0} & \cdots & \boldsymbol{x}_{i-1} & \boldsymbol{0} & \boldsymbol{x}_i \\
0 & y_1 & \cdots & 0 & y_{i-1} & 0 \\
\boldsymbol{0} & \boldsymbol{x}_1 & \cdots & \boldsymbol{0} & \boldsymbol{x}_{i-1} & \boldsymbol{0} \\
\boldsymbol{w}_k^1 & \boldsymbol{0} & \cdots & \boldsymbol{w}_k^{i-1} & \boldsymbol{0} & \boldsymbol{w}_k^i \\
* & * & \cdots & \sum_{j=1}^{i-2} \left( \boldsymbol{w}_k^{i-1\top} \boldsymbol{x}_j - y_j \right) \boldsymbol{x}_j & * & \sum_{j=1}^{i-1} \left( \boldsymbol{w}_k^{i\top} \boldsymbol{x}_j - y_j \right) \boldsymbol{x}_j \\
0 & 0 & \cdots & \frac{1}{i-2} & 0 & \frac{1}{i-1} \\
1 & 0 & \cdots & 1 & 0 & 1 \\
1 & 0 & \cdots & 0 & 0 & 0 \\
0 & 1 & \cdots & 0 & 0 & 0
\end{bmatrix}.
$$

Since the feed-forward layer is capable of approximating nonlinear functions, e.g., multiplication, the transformer layer outputs

$$\begin{bmatrix} \boldsymbol{x}_1 & \boldsymbol{0} & \cdots & \boldsymbol{x}_{i-1} & \boldsymbol{0} & \boldsymbol{x}_i \\ 0 & y_1 & \cdots & 0 & y_{i-1} & 0 \\ \boldsymbol{0} & \boldsymbol{x}_1 & \cdots & \boldsymbol{0} & \boldsymbol{x}_{i-1} & \boldsymbol{0} \\ \boldsymbol{w}_{k+1}^1 & \boldsymbol{0} & \cdots & \boldsymbol{w}_{k+1}^{i-1} & \boldsymbol{0} & \boldsymbol{w}_{k+1}^i \\ 0 & 0 & \cdots & \frac{1}{i-2} & 0 & \frac{1}{i-1} \\ 1 & 0 & \cdots & 1 & 0 & 1 \\ 1 & 0 & \cdots & 0 & 0 & 0 \\ 0 & 1 & \cdots & 0 & 0 & 0 \end{bmatrix},$$

where $\boldsymbol{w}_{k+1}^j = \boldsymbol{w}_k^j - \eta \frac{\partial \mathcal{L}}{\partial \boldsymbol{w}} \left( \boldsymbol{w}_k^j ; \boldsymbol{P}^j \right) = \boldsymbol{w}_k^j - \frac{\eta}{j-1} \sum_{h=1}^{j-1} \left( \boldsymbol{w}_k^{h\top} \boldsymbol{x}_h - y_h \right) \boldsymbol{x}_h.$ $\qquad \square$

