# OpenReview forum: "AlgoFormer: An Efficient Transformer Framework with Algorithmic Structures"
_TMLR — Accepted by TMLR_

### Review · Reviewer_K1bS · 2024-11-13

**Summary Of Contributions:**

This paper describes AlgoFormer, a variant of the looped transformer that adds a "pre-transformer" layer and a "post-transformer" layer. It shows theoretically that several in-context learning problems can be (approximately) expressed by AlgoFormers, and carries out experiments to validate those results.

**Audience:**

Yes

**Claims And Evidence:**

No

**Requested Changes:**

# Critical

1. Since the AlgoFormer is not really that different from a looped transformer or indeed a standard transformer, perhaps it should not be given a brand new name. And since its expressivity is essentially the same as the other two models, perhaps the title of the paper should not focus on expressivity.

2. There are quite a few places where the constructions are approximate, not exact, and where dependencies on the input (length) come in. The statements of the theorems should be precise about the goodness of the approximation and about the dependencies on the input.

3. In both the theoretical constructions and experiments, please clarify what is fixed by hand, what is learned in training, and what is learned-in-context.

# Non-critical

## Terminology
- The word "block" is often used to refer to a single self-attention layer and FFNN; your use of the word "block" to refer to the whole network was confusing at first.
- The word "regularizing" makes me think of regularization as in L2 regularization, so I'd suggest not using this word.

## Organization
- The definition of AlgoFormer is buried in the Motivation section.
- Why are there new results in the Discussion section?
- The training algorithm is deferred to the Experiments section, which is a common choice, but since this seems to be a key part of the method, I'd suggest presenting it earlier.

## Experiments
- \Delta T is introduced without being defined. Is it just the maximum number of layers (iterations) that are included in the loss function?
- Please explain how the looped transformer and AlgoFormer are used for MT. Is it an encoder-decoder, and if so, is the loop in the encoder, the decoder, or both?
- It seems strange to me to use cross-entropy to evaluate machine translation.

**Strengths And Weaknesses:**

Yang et al (2024) previous showed experimentally that “vanilla” looped transformers can learn to do linear regression in-context. This paper provides some theoretical constructions to back up these findings, and also extends them to more difficult variations of the problem.

# Expressivity

The AlgoFormer differs from the "vanilla" looped transformer only by the addition of a layer at the bottom and the top. Additionally, since the number of iterations of the loop is fixed (T), the AlgoFormer really amounts to a standard transformer with T+2 layers, where the parameters of layers 2 through T+1 are tied.

On page 4, some differences with "vanilla" looped transformers are listed: AlgoFormer is claimed to be "closer to the transformer in real applications," but this isn't really explained or justified as far as I could see; it's claimed that the vanilla model uses "task-specific knowledge...like token order switching" whereas AlgoFormer is "task-independent," but again I didn't really understand this claim.

At the end of section 2, it's stated that AlgoFormer is not actually more expressive than the "vanilla" looped transformer, but it can use fewer parameters. I believe this is correct, but this is a statement about parameter efficiency, not expressivity.

# Section 3.1/A.1

The position embedding uses "quasi-orthogonal" vectors with dimension O(log N) (Lemma A.1). Since N is the number of data points, doesn't this contradict the claim on page 17 that the transformer is independent of the data?

Lemma A.2 is similar to proofs by van Oswald et al, and Akyurek et al, but uses softmax instead of linear attention. As a result, it's only approximate, making use of the fact that exp x \approx 1 + x. What value of C is needed to make the approximation in the second equation of page 16 work? Doesn't C have to depend on the x's, and wouldn't that contradict the claim on page 17 that the transformer is independent of the data?

# Learning setup

I was a little confused about what is fixed by hand, what is learned in training, and what is learned-in-context.

I gathered that the representation function \Phi^* is simulated by the transformer but *not* learned in-context. However, the statement of Theorem 3.1 does not make this clear at all. On the contrary, it says that the transformer “fit[s] the representation and appl[ies] gradient descent,” making the two sound like one single learning process. Unless I’ve deeply misunderstood, the statement of this and other theorems needs to be corrected.

Similarly, in the experiments, it was not clear which parameters are fixed and which are learned. Did you fix the parameters as in the construction of Theorem 3.1, while learning the parameters for the FFNs that fit \Phi^*? Or did you learn all the parameters?

---

> ### Author Response · Authors · 2024-11-28
> **Response to Reviewer K1bS (1)**
>
> We thank the reviewer for constructive comments, which have helped us significantly improve the quality of our paper. Below, we address the comments and questions raised. All revised or modified sections are highlighted in blue in the updated version of the paper for clarity.
>
> **Comment 1.** The AlgoFormer differs from the "vanilla" looped transformer only by the addition of a layer at the bottom and the top.
>
> **Answer 1.** While it is true that a single-layer transformer may suffice in scenarios with simple preprocessing and postprocessing requirements, the design of AlgoFormer is more versatile and not constrained to such cases. Our primary claim is that the efficient and effective design of transformer architectures can be guided by mimicking the structure of algorithms. We have included the insight written in blue in Section 2.
>
> For instance, when prior knowledge suggests that a task involves complex processing and multiple iterative loops, the AlgoFormer architecture can be designed to include multi-layer pre-transformers, multiple looped transformers, and even nested looped transformers, to accommodate these requirements. This adaptability is demonstrated in Theorems 3.1 and 3.3, where the pre-transformer is not limited to a single layer due to the complexity of the task being addressed.
>
> In practical applications, the number of layers and their configurations are determined by the specific requirements of the task at hand. The flexibility in the design allows AlgoFormer to effectively handle a wide range of computational challenges.
>
> **Comment 2.** AlgoFormer is claimed to be "closer to the transformer in real applications," but this isn't really explained or justified as far as I could see; it's claimed that the vanilla model uses "task-specific knowledge...like token order switching" whereas AlgoFormer is "task-independent," but again I didn't really understand this claim.
>
> **Answer 2.** We acknowledge that the original phrasing may have been unclear, and we clarify and improve our explanation. We propose to remove the ambiguous sentences and rewrite them as follows (highlighted in blue in Section 2):
> In contrast to Giannou et al. (2023) and Yang et al. (2024), which primarily focus on tasks solvable by iterative algorithms, our approach introduces additionally transformer components (e.g., pre- and post-transformers) for processing. These components are crucial for addressing the processing needs of real-world applications. This design makes the AlgoFormer capable of representing more complex algorithms and solving challenging tasks more efficiently. Additionally, one of the core insights of our work is that transformer architectures can be designed more efficiently, flexibly, and diversely by leveraging prior knowledge and the pre-defined structure of potential algorithms. This approach enables AlgoFormer to generalize across a broader range of applications while maintaining high efficiency and adaptability.
>
>
> **Comment 3.** At the end of section 2, it's stated that AlgoFormer is not actually more expressive than the "vanilla" looped transformer, but it can use fewer parameters. I believe this is correct, but this is a statement about parameter efficiency, not expressivity.
>
> **Answer 3.** We agree with the reviewer that the "vanilla" looped transformer is sufficiently expressive, provided it has a sufficient number of loop inner layers and heads. At the end of Section 2, we present a specific example where a looped transformer with additional heads can represent a simple AlgoFormer composed of one-layer pre-, looped, and post-transformers. This example highlights the relationship between expressivity and efficiency.
>
> The focus in this section is on efficiency, which is why the subtitle reads "Expressiveness and efficiency." We specifically mention that "AlgoFormer is more efficient for representation." While both architectures are expressive, AlgoFormer achieves this with fewer parameters, highlighting its efficiency in representation without compromising on expressivity. This dual focus on expressiveness and efficiency is a key feature of our design. We changed the subtitle to “Efficiency” as suggested by the reviewer.

---

> ### Author Response · Authors · 2024-11-28
> **Response to Reviewer K1bS (2)**
>
> **Comment 4.** The position embedding uses "quasi-orthogonal" vectors with dimension O(log N) (Lemma A.1). Since N is the number of data points, doesn't this contradict the claim on page 17 that the transformer is independent of the data?
>
> **Answer 4.** We appreciate the reviewer’s comments. To clarify, in training tasks, we typically assume a maximum input length, which is represented by N and is fixed. The claim that the transformer is "independent of the data" refers to the fact that the construction of the model parameters does not depend on the exact values of the input data.
>
> To make this clearer, we have highlighted the maximal sample size N in blue in Section 3.1 of the revised manuscript. This adjustment ensures consistency between the usage of N and our claim regarding data independence.
>
>
> **Comment 5.** What value of C is needed to make the approximation in the second equation of page 16 work? Doesn't C have to depend on the x's, and wouldn't that contradict the claim on page 17 that the transformer is independent of the data?
>
> **Answer 5.** If C is sufficiently large, the error introduced by the softmax approximation can be made arbitrarily small. In our analysis, we naturally assume that the training data are bounded, meaning that C does not depend on the exact value of the data. We appreciate the reviewer pointing out the lack of rigor in our presentation. We assume that all input data are bounded, ensuring that C does not depend on the specific values of the input data, which has been highlighted in blue in Section 3.1.
>
> **Comment 6.** I was a little confused about what is fixed by hand, what is learned in training, and what is learned-in-context.
>
> **Answer 6.**
> Thank you for your comment. Below is a detailed clarification:
>
> **Trainable Components:**
>
> All trainable parameters in each component of AlgoFormer (including pre-, looped, and post-transformers) are all trainable. Specifically, as shown in Equation (1), the value (W_V), key (W_K), query (W_Q), weight (W_1, W_2), and bias (b_1, b_2) matrices for each layer and each head are trainable. In Equation (6), \Theta denotes the set of all those parameters for all components and is trainable.
>
> **Fixed Components:**
>
> (1) The starting step (T_0), the overall loop count (T), the number of training samples (N), and the architecture of the transformers (e.g., number of layers and heads for each component of AlgoFormer) are fixed by hand.
>
> (2) The sampling distribution for input data, the representation function \Phi^{\star} and the distributions for Leaky-ReLU networks for generating CoT sequences are fixed for each task.
>
> **Learned in Training vs. In-Context Learning:**
>
> (1) The representation function \Phi^{\star} is learned during training but is not learned in-context.
>
> (2) The linear weights A in the Regression and AR(q) tasks, as well as the leaky ReLU MLP used in CoT tasks, are learned in-context.
>
> **To make all those clear to readers, we elaborate on them in blue. For example, we include:**
>
> (1) For Regression in Section 3.1: Here, the weight matrix $A$ varies across different sequences of in-context samples but remains constant within a single sequence. The representation function $\Phi^{*}$, on the other hand, is fixed across all samples of sequences.
>
> (2) For AR(q) in Section 3.2: $\Phi^{*}\left(\cdot\right)$ is a fixed representation function (e.g., we take the L-layer MLPs), and the weight $A$ vary from different sequences but remains constant within a single sequence.
>
> (3) For CoT with MLPs in Section 3.3: The implicit $L$-layer MLP remains the same within a single sequence but varies across different sequences, and it is learned in-context.
>
> We also have highlighted what’s learned and learned in-context in the experimental part (Section 5.1).

---

> ### Author Response · Authors · 2024-11-28
> **Response to Reviewer K1bS (3)**
>
> **Comment 7.** Since the AlgoFormer is not really that different from a looped transformer or indeed a standard transformer, perhaps it should not be given a brand-new name. And since its expressivity is essentially the same as the other two models, perhaps the title of the paper should not focus on expressivity.
>
> **Answer 7.** We appreciate the reviewer’s understanding of the model and their thoughtful comments. We chose the name "AlgoFormer" to emphasize the insight that efficient transformer architectures can be designed by leveraging prior knowledge of the algorithmic structure.
>
> While the AlgoFormer shares similarities with the looped transformer, it is not restricted to being a two-layer architecture. Instead, it can incorporate multi-processing, multi-looped and even nested looped layers, depending on the specific prior knowledge about the algorithmic structure. This flexibility allows it to adapt to different problem settings while maintaining efficiency and effectiveness.
>
> Regarding the title, we will consider rephrasing it to better reflect the unique contributions of the work, ensuring it aligns with the paper's focus on the integration of algorithmic principles and structures into transformer design for improving the efficiency. For example, **“AlgoFormer: Designing Efficient Transformer Framework with Algorithmic Structures”.**
>
>
> **Comment 8.** There are quite a few places where the constructions are approximate, not exact, and where dependencies on the input (length) come in. The statements of the theorems should be precise about the goodness of the approximation and about the dependencies on the input.
>
>
> **Answer 8.** We appreciate the reviewer’s observation. We acknowledge that similar issues arise in prior work, such as Giannou et al. (2023), due to the linearization of the softmax function. To address this, we have included explicit statements in each theorem to clarify these approximations.
> For example, The emulation of each step is not exact, as there is some error introduced in each step. However, the error can be made arbitrarily close to zero by increasing the temperature of the softmax and adjusting another free parameter, neither of which affects the size of the network.
>
>
> **Comment 9.** In both the theoretical constructions and experiments, please clarify what is fixed by hand, what is learned in training, and what is learned-in-context.
>
> **Answer 9.** We revised accordingly, highlighted in blue in the modified version.
>
> (1) In Section 3.1: Here, the weight matrix $A$ varies across different sequences of in-context samples but remains constant within a single sequence, and is learned in-context. The representation function $\Phi^{*}$, on the other hand, is fixed across all samples of sequences and is learned during training.
>
> (2) In Section 3.2: Our goal is to learn   $\Phi^{*}\left(\cdot\right)$ during training and to perform in-context learning of the weight matrix $A$.
>
> (3) In Section 3.3: The implicit $L$-layer MLP remains the same within a single sequence but varies across different sequences, and it is learned in-context.
>
> (4) In Section 5.1: For Regression and AR(q): Here, our target is to learn the representation function $\Phi^{*}(\cdot)$ during training and to perform in-context learning of the weight matrix $A$.
> For CoT with MLPs: Here, our target is to perform in-context learning of the generator function (i.e., the 6-layer leaky ReLU MLP).
>
>
> **Comment 10.** The word "block" is often used to refer to a single self-attention layer and FFNN; your use of the word "block" to refer to the whole network was confusing at first.
>
> **Answer 10.** Thank you for pointing this out. To avoid confusion, we will use the term “framework” instead of "block" for clearer presentation throughout the paper.
>
>
> **Comment 11.** The word "regularizing" makes me think of regularization as in L2 regularization, so I'd suggest not using this word.
>
> **Answer 11.** You are correct that the term "regularizing" may be confused with traditional forms of regularization, such as L2 regularization. In our paper, we use "regularizing" to refer specifically to a type of structure regularization, where the design enforces certain architectural constraints or properties in the network. To avoid confusion, we will clarify this in the text and ensure the term is used consistently to reflect its intended meaning in the context of our work. We replace instances of "regularizing" with "structure regularization" or explicitly define it where first introduced.

---

> ### Author Response · Authors · 2024-11-28
> **Response to Reviewer K1bS (4)**
>
> **Comment 12.** The definition of AlgoFormer is buried in the Motivation section.
>
> **Answer 12.** To improve clarity and accessibility, we change the title of Section 2 to “Motivation and Proposed Method.” This adjustment will make it more straightforward for readers to locate and understand the definition of AlgoFormer within the context of the paper. We also moved the training strategy to Section 2.3.  We have included more details and discussion of AlgoFormer in Section 2.2. These modifications are highlighted in blue.
>
>
>
> **Comment 13.** Why are there new results in the Discussion section? The training algorithm is deferred to the Experiments section, which is a common choice, but since this seems to be a key part of the method, I'd suggest presenting it earlier.
>
> **Answer 13.** We change the section title to “Extensions and Further Analysis” for some new results. We move the training methodology to Section 2.3 in the proposed method part. We also included additional discussions in Section 6. These modifications are highlighted in blue.
>
>
>
> **Comment 14.** $\Delta T$ is introduced without being defined. Is it just the maximum number of layers (iterations) that are included in the loss function?
>
> **Answer 14.** Delta T is indeed the difference between the maximal loop iteration T and the initial step T_0, representing the maximum number of iterations (or layers) included in the loss function. Its primary purpose is to train a more robust transformer. While our main focus is on the final output of the AlgoFormer, incorporating the loss from T_0 to T ensures that the transformer faithfully applies iterative algorithms and generalizes beyond the number of training loops. This approach helps enhance the transformer's capability for tasks requiring a varying number of iterations.
>
> We clarify this definition in Section 2.3 of the revised manuscript for better readability.
>
>
> **Comment 15.** Please explain how the looped transformer and AlgoFormer are used for MT. Is it an encoder-decoder, and if so, is the loop in the encoder, the decoder, or both?
>
> **Answer 15.** Each layer of the looped transformer and AlgoFormer operates as a decoder-only transformer, following the trend established by the success of decoder-only networks, such as GPT. Some of our results can also be extended to decoder-based transformers, as demonstrated in Theorem 4.2.
>
> We elaborate on the setting of the German-to-English translation task. In this task, the input German text is treated as a prefix, while the output English text is generated autoregressively using the decoder-based looped transformer and AlgoFormer. This design allows the models to process translation tasks effectively within a decoder-only framework. We included the details in Section 5.5.
>
>
> **Comment 16.** It seems strange to me to use cross-entropy to evaluate machine translation.
>
> **Answer 16.** As suggested, we included BLUE as the performance criteria. We also observe that AlgoFormer achieves the highest BLUE scores.

---

### Review · Reviewer_jrwC · 2024-11-20

**Summary Of Contributions:**

A new Transformer-based model, called AlgoFormer, is introduced for improving algorithmic capabilities of Transformers. AlgoFormer adds pre-and post-Transformer blocks to the previously proposed looped Transformer structure to enhance expressive power.

**Audience:**

Yes

**Claims And Evidence:**

Yes

**Requested Changes:**

Critical for acceptance:
+ Add specific details on how training and inference are performed. (a) Are the pre-, looped- and post-Transformer blocks jointly trained end-to-end? Or are they trained on different inputs and/or with different objective functions? (b) The dashed arrows in Figure 1 are very confusing. Do they indicate that new tokens are introduced after each block? For instance, after the pre-Transformer blocks are completed, are there new tokens that are added to represent the loop function before feeding as input to the looped Transformer block? (c) How exactly is the prompt constructed for a given input in the experiments? Does it follow the P matrix defined in section 3.1? If so, is there any intuition behind this? Is the same prompt generation strategy used in all baselines?
+ Discuss the training and inference time overheads introduced by AlgoFormer over looped transformer. Do the pre- and post-Transformer blocks add substantial overheads? Or is the runtime dominated by the looped Transformer blocks?
+ Discuss the scalability of the proposed method to more complex tasks. For instance, can AlgoFormer solve (or can be trivially modified to solve) problems that requires the use of nested loops? Or are there any assumptions made on the structure of tasks/algorithms when computing expressive power?

Would strengthen the paper:
+ A figure (using a simple toy example) showing the end-to-end flow from prompt generation to output production would be very helpful to understand the inner workings of the proposed method
+ A discussion on the limitations of the proposed method and the types of tasks it can solve

**Strengths And Weaknesses:**

Strengths:
+ The intuition behind the proposed method is clearly explained and makes sense to me. In particular, the section linking the proposed model design to how humans solve these problems provides a clear intuitive basis for why the model works better than the standard looped transformers.
+ The use of toy examples to empirically demonstrate the advantages of the proposed method (in addition to the theoretical analysis) demonstrates that this method can indeed be useful in real-world scenarios.

Weaknesses:
+ Key implementation details are missing, making it difficult to understand how everything works end-to-end, and the additional complexity introduced by AlgoFormer over the looped Transformer.
+ Scalability of the proposed method to more difficult tasks is unclear

---

> ### Author Response · Authors · 2024-11-28
> **Response to Reviewer jrwC (1)**
>
> We thank the reviewer for constructive comments, which have helped us significantly improve the quality of our paper. Below, we address the comments and questions raised. All revised or modified sections are highlighted in blue in the updated version of the paper for clarity.
>
> **Comment 1.** Add specific details on how training and inference are performed. (a) Are the pre-, looped- and post-Transformer blocks jointly trained end-to-end? Or are they trained on different inputs and/or with different objective functions?
>
> **Answer 1.** The pre-, looped-, and post-Transformer blocks are jointly trained end-to-end. We denote the set of all trainable parameters as $\Theta$, and minimize the training loss defined in Equation (3). To clarify this in the paper, we explicitly state that “all transformer modules are jointly trained end-to-end.”
> Compared to the standard training strategy, our method builds on the approach proposed in Yang et al. (2024), where the loss function incorporates different loop numbers  (controlled by T and \Delta T). This strategy ensures robust training by allowing the model to generalize across varying numbers of iterations, improving both training and inference performance.
>
> **Comment 2.** (b) The dashed arrows in Figure 1 are very confusing. Do they indicate that new tokens are introduced after each block? For instance, after the pre-Transformer blocks are completed, are there new tokens that are added to represent the loop function before feeding as input to the looped Transformer block?
>
> **Answer 2.**
> The dashed arrows in Figure 1 do not indicate the introduction of new tokens. All input to the next layer is the output from the previous layer. Instead, they imply that the operations in the algorithm correspond to specific transformer modules. For example, “statement 1” in the algorithm serves as preprocessing, which is mapped to the pre-transformer module in AlgoFormer. Similarly, “statement 2,” which represents an iterative algorithm, corresponds to the looped transformer module, and “statement 3” represents postprocessing, which is handled by the post-transformer module.
>
> The purpose of the dashed arrows is to illustrate how each part of the algorithm aligns with a specific transformer component. If we have prior knowledge of the algorithm structure, we can design AlgoFormer to mirror that structure. This framework allows AlgoFormer to efficiently and flexibly mimic the structure of the underlying algorithm and leverage it for task-specific applications. For example, if we have prior knowledge that the algorithm for the task involves nested loops, then we can similarly design the AlgoFormer with nested looped transformer module to represent such algorithm.
>
>
> **Comment 3.** (c) How exactly is the prompt constructed for a given input in the experiments? Does it follow the P matrix defined in section 3.1? If so, is there any intuition behind this? Is the same prompt generation strategy used in all baselines?
>
> **Answer 3.** Yes, the input prompt is constructed strictly following the P matrix defined in Section 3.1. This is a common setup, adopted in some previous papers (e.g., Yang et al. 2024, Bai et al. 2023, Akyürek et al., 2023; Garg et al., 2022). The P matrix in Section 3.1 is specifically designed for regression tasks. For AR(q), the P matrix is slightly different, and we provide the detailed formulation in Section 3.2. These modifications are highlighted in blue.
>
> The formulation of the P matrix may vary slightly depending on the task, but the underlying intuition remains the same. For a given sequence of in-context samples {x_1, y_1, x_2, y_2, … , x_N, y_N}, the observations x_i and the corresponding responses y_i are not in the same dimension. To ensure that all input tokens share the same dimension, we construct the P matrix, where columns (tokens) are in the same dimension.  The P matrix organizes the tokens such that they are in the same dimension while retaining the ability to distinguish between observations x_i and responses y_i through their positions.
>
> For AR(q) and CoT-MLPs tasks, we adopt similar prompt construction strategies. Although the specific structure of the P matrix may be adapted for different tasks, the approach ensures that the input format aligns with the requirements of the transformer and supports task-specific adaptations.
>
> To enhance clarity for readers, we have included the following statement in the Remarks of Section 3.1: “Due to the differing dimensions of the input x and its corresponding label y,  zero padding is incorporated to reshape them into vectors of the same dimension.”

---

> ### Author Response · Authors · 2024-11-28
> **Response to Reviewer jrwC (2)**
>
> **Comment 4.** Discuss the training and inference time overheads introduced by AlgoFormer over looped transformer. Do the pre- and post-Transformer blocks add substantial overheads? Or is the runtime dominated by the looped Transformer blocks?
>
> **Answer 4.** The additional training and inference time introduced by AlgoFormer compared to the looped transformer largely depends on the size of the pre- and post-transformer modules. If the pre- and post-transformer modules are similar in size to the looped transformer, the additional computational cost becomes negligible, especially when the loop iteration T is much greater than the number of layers in these modules.
>
> For example, in our experiments, we set the pre-, looped, and post-transformer layers to 1, and T=20 during training. In this case, the additional computation introduced by the pre- and post-transformer modules is approximately 1/10 of the total computation. During inference, where T is typically much larger than in training (e.g., T=200 loop iterations in Figure 3), the additional computational overhead is reduced further, to about 1/100.
>
> Thus, we believe that for most scientific and language tasks, when using AlgoFormer, the runtime is generally dominated by the looped transformer. The overhead introduced by the pre- and post-transformer modules remains minimal in comparison.
>
> We have included this discussion in the experimental sections (Section 5.6) of the revised manuscript to provide further clarity. These modifications are highlighted in blue.
>
>
> **Comment 5.** Discuss the scalability of the proposed method to more complex tasks. For instance, can AlgoFormer solve (or can be trivially modified to solve) problems that require the use of nested loops? Or are there any assumptions made on the structure of tasks/algorithms when computing expressive power?
>
> **Answer 5.**
> We appreciate the reviewer for raising this insightful question.
>
> While the examples in this paper focus on the simplest case of pre-, looped-, and post-transformers, AlgoFormer is not restricted to this three-part structure. Instead, we propose a highly flexible transformer framework that can be adapted to align with more complex algorithmic structures, such as nested loops. For tasks that inherently involve nested loops, AlgoFormer can be trivially modified to incorporate nested looped transformers. This modification would involve adapting the architecture to match the nested loop structure of the algorithm.
>
> However, to stabilize training in such cases, the training strategy outlined in Section 2.3 would need to be adjusted. Specifically, the loss objective would need to include loop iterations from all levels of nested loops. While this approach introduces additional computational overhead, it significantly enhances the expressiveness and capability of the transformer framework, enabling it to represent and conduct more complex algorithms.
>
> This paper serves as a starting point for exploring how algorithmic structures can inform the design of transformer framework. It primarily provides insight into designing efficient and flexible transformer frameworks based on algorithmic structures (including nested loops). While the implementation of AlgoFormer for representing more complex algorithms, such as those involving nested loops, is beyond the scope of this paper, where we provide a key concept, we see this as a promising avenue for future research.
>
> We have included this discussion in the paper (Section 6) to clarify the scalability and flexibility of AlgoFormer.

---

> ### Author Response · Authors · 2024-11-28
> **Response to Reviewer jrwC (3)**
>
> **Comment 6.** A figure (using a simple toy example) showing the end-to-end flow from prompt generation to output production would be very helpful to understand the inner workings of the proposed method.
>
> **Answer 6.** In Figure 3, we provide insights into the behavior of AlgoFormer during inference. Although we train AlgoFormer with T=20 loop iterations, we apply inference with significantly longer loop iterations (e.g., T=200),
> As T increases, AlgoFormer demonstrates improved performance and eventually stabilizes (i.e., converges). This behavior suggests that the inner looped transformers are effectively performing iterative computations that converge to a stable solution.
>
> We realize that the discussion of this behavior in Figure 3 was missing in the original paper, and we have included it in the revised version located at the end of Section 5.3.
>
> **Comment 7.** A discussion on the limitations of the proposed method and the types of tasks it can solve.
>
> **Answer 7.** We include the discussion of potential challenges and limitations in practical applications in Section 6 (Conclusion and Discussion). These modifications are highlighted in blue.

---

> > ### Comment · Reviewer_jrwC · 2024-12-15
> > **Response to rebuttal**
> >
> > Thanks to the authors for the responses and revisions. My main concerns have been addressed. One minor point -- I still believe that a figure using a toy example (like a very simple 'for' loop) to show the end-to-end flow from prompt generation to the output (i.e., how the the input is processed and the output is generated) will help readers understand the full working of AlgoFormer. This is especially important since the details of the end-to-end flow are scattered in the different sections, making it difficult to connect everything together in the first pass through the paper.

---

> > > ### Author Response · Authors · 2024-12-15
> > >
> > > We sincerely thank the reviewer for their thoughtful feedback and for highlighting the importance of a figure to illustrate the end-to-end process of AlgoFormer. We understand that such a visualization would make it easier for readers to grasp the full workflow from prompt generation to output, especially given that the relevant details are currently distributed across different sections of the paper.
> > >
> > > To address this point, let us take linear regression $w^{* \top} x$ as an example. Due to the training strategy described in Equation (3), AlgoFormer focuses primarily on the final prediction and progressively and iteratively computes $w_{k}^{\top} x_{\text{test}}$, where $k$ is the loop number, rather than explicitly computing the intermediate linear weight
> > >  $w_{k}$. In Figure 3, we illustrate this by plotting the trajectory of $w_{k}^{\top} x_{\text{test}} – w^{* \top} x_{\text{test}} $ to measure the closeness of $ w_{k}$ to $w^{*}$. This approach aligns with related works [1–4], where the intermediate computation of $w_{k}$ is not explicitly conducted by transformers. Instead, the Transformer directly computes $w_{k}^{\top} x_{\text{test}}$, making the explicit visualization and computation of $w_{k}$ itself less straightforward under the current training strategy (Equation (3)).
> > >
> > > We find the reviewer’s suggestion both interesting and valuable for future research. A potential direction could involve designing a more powerful training strategy that supervises AlgoFormer to explicitly represent some pre-defined intermediate algorithmic steps. Although we theoretically demonstrate that AlgoFormer has the capacity to represent algorithms, this aspect remains unexplored under the current framework. We will include a discussion of this idea in the paper as a potential avenue for future work.
> > >
> > >
> > > [1] Akyürek, Ekin, Dale Schuurmans, Jacob Andreas, Tengyu Ma, and Denny Zhou. " What learning algorithm is in-context learning? Investigations with linear models." In The Eleventh International Conference on Learning Representations. 2022.
> > >
> > > [2] Bai, Yu, Fan Chen, Huan Wang, Caiming Xiong, and Song Mei. "Transformers as statisticians: Provable in-context learning with in-context algorithm selection." Advances in Neural Information Processing Systems 36 (2024).
> > >
> > > [3] Li, Yingcong, Muhammed Emrullah Ildiz, Dimitris Papailiopoulos, and Samet Oymak. "Transformers as algorithms: Generalization and stability in in-context learning." In International Conference on Machine Learning, pp. 19565-19594. PMLR, 2023.
> > >
> > > [4] Guo, Tianyu, Wei Hu, Song Mei, Huan Wang, Caiming Xiong, Silvio Savarese, and Yu Bai. "How Do Transformers Learn In-Context Beyond Simple Functions? A Case Study on Learning with Representations." In The Twelfth International Conference on Learning Representations. 2023.

---

### Review · Reviewer_E2zR · 2024-11-21

**Summary Of Contributions:**

This paper introduces AlgoFormer, a novel transformer architecture designed to learn and implement algorithms through a structured approach combining pre-processing, iterative optimization, and post-processing steps. The authors demonstrate both theoretically and empirically that AlgoFormer can effectively learn and execute various algorithms, particularly for tasks involving optimization.

**Audience:**

Yes

**Claims And Evidence:**

Yes

**Requested Changes:**

I am proposing some suggestions here, with the understanding that this paper is primarily a theoretical work, but I think these would make it a better paper:

1. Include more real-world applications beyond German-English machine translation
2. Include analysis of training stability and computational cost
3. Discuss potential challenges in practical applications

Writing:
In general, the writing could use some polishing, some paragraphs could've been better.

"However, it is still unverified that the standard multi-layer transformer is exactly performing algorithms."

"A One-layer transformer is mathematically formulated as:" -> "A one-layer..."

**Strengths And Weaknesses:**

**Strengths**

1. Provides rigorous theoretical analysis with detailed proofs for the expressive power of AlgoFormer
2. Demonstrates capability to implement both first-order (gradient descent) and second-order (Newton's method) optimization algorithms
3. Novel three-part structure (pre-transformer, looped transformer, post-transformer) mimics human-designed algorithms

**Weaknesses**

1. While the German-English translation task provides real-world validation, additional practical applications would strengthen the paper.
2. The three-part structure, while theoretically elegant, may introduce additional complexity in training
3. Continuing the last point, how do the authors see AlgoFormer being used in real-world scenarios, and how would one decide on the architecture (how big is pre-processing/looped/post-processing transformer), and how many loops to execute?

---

> ### Author Response · Authors · 2024-11-28
> **Response to Reviewer E2zR (1)**
>
> We thank the reviewer for constructive comments, which have helped us significantly improve the quality of our paper. Below, we address the comments and questions raised. All revised or modified sections are highlighted in blue in the updated version of the paper for clarity.
>
> **Comment 1.** While the German-English translation task provides real-world validation, additional practical applications would strengthen the paper.
>
> **Answer 1.** We are currently conducting additional experiments on real-world language tasks using AlgoFormer, such as AG News classification and numerical addition tasks. Due to time constraints, the final results are still in progress, but we aim to include these results as soon as possible before the deadline.
>
>
> **Comment 2.** The three-part structure, while theoretically elegant, may introduce additional complexity in training.
>
> **Answer 2.** We agree with the reviewer. The training strategy (now presented in Section 2.3) is more complicated than standard transformer. The three-part structure acts as a structure regularization for transformers in conducting certain “algorithms”. Therefore, we think it is reasonable that the introduced method may bring additional complexity in training. Also, the additional complexity (both for computer programming and computational costs) is affordable.
>
> The training strategy is placed in Section 2.3. The additional computational costs of AlgoFormer compared with the standard transformer and the vanilla looped transformer are discussed in Section 5.6. These modifications are highlighted in blue.
>
>
>
> **Comment 3.** Continuing the last point, how do the authors see AlgoFormer being used in real-world scenarios, and how would one decide on the architecture (how big is pre-processing/looped/post-processing transformer), and how many loops to execute?
>
> **Answer 3.**
> The selection of hyperparameters, such as the number of layers for each module, layer width, and other architectural details, is always critical for deep learning models. Scaling laws and generalization theory suggest that larger models trained with more data tend to perform better. Therefore, the specific architecture,  such as the number of layers in the pre-processing, looped, and post-processing modules, as well as hidden dimensions, should be set based on the complexity of the task and the availability of training data.
>
> The loop count, \Delta T, is another important factor. While increasing \Delta T contributes to stable training and enhances the model’s generalization capabilities, it also increases computational costs during training. Thus, there is a trade-off between model size, loop count, computational efficiency, and overall performance.
>
> This paper serves as a starting point for exploring how algorithmic structures can guide the design of transformer framework. Most of the examples in this work focus on scientific computing and illustrative cases (e.g., regression, AR(q), and CoT with MLPs). We also extend our experiments to a language task, specifically German-English translation, where we observe superior performance with the proposed method. We believe that the AlgoFormer framework holds significant potential and can be scaled up  for real-world language tasks and other applications, in the future.
>
> **Comment 4.** Include analysis of training stability and computational cost.
>
> **Answer 4.**
> The stability of AlgoFormer is reflected in its ability to generalize from the loop iterations T used during training to significantly longer loop iterations during inference.
> For example, in Figure 3, we train the model with T=20, while conduct the inference with significantly longer loop iterations T=200. As T increases beyond the training length, AlgoFormer demonstrates improved performance and eventually stabilizes (i.e., converges). This behavior suggests that the inner looped transformers are effectively performing iterative computations that converge to a stable solution. The discussion has been included in Section 5.3.
>
> Regarding computational costs, the additional overhead of AlgoFormer compared to the standard transformer and the vanilla looped transformer depends on the specific structure of the framework. We analyze these computational costs for both training and inference in Section 5.6. The discussion includes an evaluation of the pre- and post-transformer modules and their impact on computational complexity, showing that the overhead is mild in our experimental settings.

---

> ### Author Response · Authors · 2024-11-28
> **Response to Reviewer E2zR (2)**
>
> **Comment 5.** Discuss potential challenges in practical applications.
>
> **Answer 5.** We have included the discussion of potential challenges in practical applications in Section 6 (Conclusion and Discussion). These modifications are highlighted in blue.
>
>
>
> **Comment 6.** Writing: In general, the writing could use some polishing, some paragraphs could've been better.
>
> **Answer 6.** Thanks for suggestion. We polish the paper as suggested.

---

> ### Author Response · Authors · 2024-12-02
> **Response to Reviewer E2zR (3) Additional real-world applications**
>
> Thanks again to the reviewer for suggesting additional experiments to evaluate AlgoFormer on real-world applications. We are making every effort to conduct more experiments during the rebuttal period to address the reviewers’ and AE's concerns and improve the paper. Currently, we are exploring more language tasks and datasets. Here, we report some additional experiments on the text classification task using various datasets (AG News, IMDB, DBPedia, Yelp Review, and Yahoo News), and this new section has been incorporated into Section 5.5 of the revised manuscript.
>
>
>
> We also implement the proposed AlgoFormer on text classification tasks using various datasets to evaluate its performance on a real-world language application. The experimental setup includes a standard Transformer with 12 layers, 2 attention heads, a feature dimension of 64, and a learning rate of 1e-3. The pre-, looped, and post-transformers are all implemented as single layers with one attention head, with $(T,\Delta T)=(10,10)$. The input news text data is processed using encoder-based transformers, and the output classification is generated. Classification accuracy, where higher accuracy indicates better performance, is used as the evaluation metric. The results are summarized in the table below:
>
> **Table**. Accuracy (%) of Transformer models on text classification across different datasets.
> | Model         | Standard | Looped | AlgoFormer|
> |--------------------|--------------|------------|----------------|
> | AG News  | 92.07        | 91.04      | **97.92**     |
> | IMDB  | 75.00        | **77.50**      | **77.50**     |
> | DBPedia  | **99.21**       | 99.11      | 98.21  |
> | Yelp Review  | **66.25**      |  53.57     | 57.50     |
> | Yahoo News  | 73.96        |  69.79      | **81.25**     |

---

### Author Response · Authors · 2025-01-02

Dear Reviewers and Action Editor,

We sincerely thank you for your time, effort, and constructive comments, which have been invaluable in enhancing the quality of our paper.


Best regards,

The Authors

---

### Decision · Action_Editor_2GiE · 2024-12-25

**Recommendation:** Accept with minor revision

**Comment:**

Reviews unanimously agree that the paper is above acceptance criteria, albeit somewhat borderline.

Reviewer K1bS and jrwC raised many points which I agree with, mainly on the imprecise language of their architecture contributions:
  * What are the differentiating mechanics between the "pre", "loop", and "post" layers? Are these layers somehow trained differently (e.g. different loss functions) or have different architecture hyperparameters (e.g. different numbers of heads)? Or do the "loop" layers all share the same set of weights?
    * This absolutely needs to be made clear in the paper. At the moment, the authors' post-rebuttal explanations still haven't satisfactorily resolved this, and it can appear as if the paper is introducing additional novelty when none exists. If they are purely for naming purposes, then the paper needs to reflect this as well, by stating explicitly that they are purely distinguished by name.
  * "vanilla model uses "task-specific knowledge...like token order switching" whereas AlgoFormer is "task-independent,": Unless these descriptions can be rigorously justified or explained, then they should be removed. It's not clear what any of these mean at all.

I expect that after a proper re-write of Sections 1 to 2.2, it can be made much more concise, down from 4 pages to possibly even 2.

**Audience:**

Yes, the paper's contributions for approximating gradient descent layer-by-layer may be of interest to those working in the theory of Transformers.

**Claims And Evidence:**

The paper proposes the following design of the "AlgoFormer":
* T+2 vanilla Transformer blocks stacked, with varying head counts per layer. The architecture is split split into "pre", "loop", "post" layers.
* Training loss function based on intermediate layer outputs, rather than fully end-to-end as in vanilla cases.

The paper then makes the claims, followed by evidence:
1. A single layer is able to approximate a case of gradient descent over:
    * Regression data $(x_1, y_1, x_2, y_2, ...)$ where these values are generated by a separate MLP (Appendix A1 Proof)
    * Autoregression data $(x_1, x_2, x_3, ...)$ where each $x_{i+1}$ is an autoregressive function of the previous (Appendix A2 Proof)
    * "Stateful chain of thought" where intermediate MLP hidden layer states are shown. (Appendix A3 Proof)

2. Empirically, the AlgoFormer outperforms standard Transformer / Looped Transformer variants over the above tasks.
    * Ablations are conducted by varying number of layers $T$ and heads
    * Comparisons are made against actual gradient-descent algorithms (e.g. Newton's method) with varying iteration counts

3. Finally, experiments are conducted on machine translation tasks (German to English) + Text classification to demonstrate possible real-world value.

While the paper does support its primary claims with theory and toy experiments, I want to raise the issue of clarity still (see below on my comment), which leads to the "Accept with minor revision" decision.

---

> ### Author Response · Authors · 2025-01-02
>
> Dear Action Editor,
>
> We sincerely thank the reviewers and you for your constructive comments, which have greatly contributed to improving the quality of our paper. We are grateful for recognizing the potential of our work and the development of efficient transformer frameworks in in-context algorithm learning, and for recommending the paper for acceptance.
> In response to the reviewers’ and your comments, we have made significant revisions to the paper. The key modifications are as follows:
>
> 1. Clarification of the AlgoFormer Framework:
>
> (1) To address concerns about the "pre," "loop," and "post" layers, we have explicitly clarified their distinctions in the manuscript. Specifically, we now state:
>
> "
> The looped transformer layers ($\text{TF}\_{\text{loop}}$) share the same set of weights, as they perform identical computations during each iteration. In contrast, the pre-transformer ($\text{TF}\_{\text{pre}}$) and post-transformer ($\text{TF}\_{\text{post}}$) utilize distinct weights and architectures to handle their specific computational roles.
> "
>
> after Equation (2).
>
> (2) Additionally, we have included the following after Equation (2):
>
> "
> The hyperparameters for each transformer module (e.g., number of heads, layers, and hidden dimensions) are configured based on prior knowledge of the computational complexity of the target algorithms.
> "
>
> (3) The overall design of AlgoFormer (composed of pre-, looped, and post-transformers) is based on the algorithmic structure, as illustrated in Figure 1. Moreover, if prior knowledge about algorithm structure (e.g., nested loops) is available, AlgoFormer can be adapted accordingly (e.g., by incorporating nested looped transformers), leveraging the ability of transformers to represent basic computations.
>
>
> (4) The training for pre-, looped, and post-transformers are different. To address differences in training, we elaborate on the training strategy outlined in Equation (3) in Section 2.3. Specifically, we clarify that the loss function involves pre- and post-transformers with varying numbers of loop layers (from $T_0$ to $T$) to stabilize training.
>
>
> 2. Removal of Ambiguous Statements:
>
> We have removed the statement: "vanilla model uses task-specific knowledge...like token order switching whereas AlgoFormer is task-independent," as suggested, to avoid confusion and ensure rigor.
>
> 3. Rewriting and Shortening Sections 1 to 2.2
>
> In line with the AE's suggestion, we have rewritten and shortened Sections 1 to 2.2, reducing their length from 4 pages to 2 pages for improved clarity and conciseness.
>
> 4. Theoretical Extensions:
>
> In addition to the in-context tasks analyzed in the original submission (e.g., linear regression, AR(q), and Chain-of-Thought reasoning), we also include a theoretical analysis of AlgoFormer’s capability to represent Newton’s method and its extension to decoder-only transformers.
>
> 5. Extended Experiments:
>
> To further validate the performance of AlgoFormer, we conducted additional experiments on real-world language tasks. These include machine translation and text classification (across five datasets), demonstrating the effectiveness and adaptability of AlgoFormer in diverse applications.
>
> 6. Clear Claims:
>
> In Sections 3 (theoretical analysis) and 5 (experimental evaluation), we have refined the claims regarding the in-context setting of the problems. Specifically, we clarify which parts of the problems should be learned in-context and which parts remain fixed.
>
> In Theorems 3.1, 3.2, 3.3, 4.1, and 4.2, we have added more precise statements emphasizing that the error introduced can be made arbitrarily small, controlled by the temperature parameter of the softmax function.
>
> 7. Potential Limitations of AlgoFormer
>
> We have included a discussion of the potential limitations of AlgoFormer in Section 6, highlighting areas where further research and exploration are needed.
>
> 8. Title of the Paper
>
> Finally, in alignment with Reviewer K1bS’s suggestion, we propose changing the title of the paper to:
> “AlgoFormer: An Efficient Transformer Framework with Algorithmic Structures”
> to better reflect the core motivation and contributions of our work.
>
>
> We thank you and the reviewers once again for your valuable feedback and support. Please let us know if there are any additional points to address.
>
> Best regards,
>
> The Authors